# Super-pangenome analyses highlight genomic diversity and structural variation across wild and cultivated tomato species

Ning Li[1,11], Qiang He [2,11], Juan Wang[1], Baike Wang[1], Jiantao Zhao [3], Shaoyong Huang[1,4], Tao Yang[1], Yaping Tang[1], Shengbao Yang[1], Patiguli Aisimutuola[1], Ruiqiang Xu[1,4], Jiahui Hu[1,4], Chunping Jia[1,5], Kai Ma[1], Zhiqiang Li[6], Fangling Jiang[7], Jie Gao[4], Haiyan Lan[5], Yongfeng Zhou [3], Xinyan Zhang[3], Sanwen Huang [3], Zhangjun Fei [8,9], Huan Wang [10] ✉, Hongbo Li [3] ✉ & Qinghui Yu [1] ✉

Effective utilization of wild relatives is key to overcoming challenges in genetic improvement of cultivated tomato, which has a narrow genetic basis; however, current efforts to decipher high-quality genomes for tomato wild species are insufficient. Here, we report chromosome-scale tomato genomes from nine wild species and two cultivated accessions, representative of *Solanum* section *Lycopersicon*, the tomato clade. Together with two previously released genomes, we elucidate the phylogeny of *Lycopersicon* and construct a section-wide gene repertoire. We reveal the landscape of structural variants and provide entry to the genomic diversity among tomato wild relatives, enabling the discovery of a wild tomato gene with the potential to increase yields of modern cultivated tomatoes. Construction of a graph-based genome enables structural-variant-based genome-wide association studies, identifying numerous signals associated with tomato flavor-related traits and fruit metabolites. The tomato super-pangenome resources will expedite biological studies and breeding of this globally important crop.

Tomato (*Solanum lycopersicum* L.) is among the most important vegetable crops in terms of global production (http://www.fao.org/faostat/en/#data/QCL), also serving as a classic model system for genetic, developmental and physiological studies of fleshy fruits[1]. It belongs to the genus *Solanum* in the nightshade family Solanaceae. Cultivated tomatoes have lost substantial genetic diversity owing to a domestication bottleneck and intensive artificial selection in pursuit of bigger fruits and higher yield[2], which has impeded tomato improvement.

[1]The State Key Laboratory of Genetic Improvement and Germplasm Innovation of Crop Resistance in Arid Desert Regions (Preparation), Key Laboratory of Genome Research and Genetic Improvement of Xinjiang Characteristic Fruits and Vegetables, Institute of Horticultural Crops, Xinjiang Academy of Agricultural Sciences, Urumqi, China. [2]Institute of Crop Sciences, Chinese Academy of Agricultural Sciences, Beijing, China. [3]Shenzhen Branch, Guangdong Laboratory for Lingnan Modern Agriculture, Shenzhen Key Laboratory of Agricultural Synthetic Biology, Genome Analysis Laboratory of the Ministry of Agriculture and Rural Affairs, Agricultural Genomics Institute at Shenzhen, Chinese Academy of Agricultural Sciences, Shenzhen, China. [4]College of Horticulture, Xinjiang Agricultural University, Urumqi, China. [5]College of Life Science and Technology, Xinjiang University, Urumqi, China. [6]Adsen Biotechnology Co., Ltd., Urumqi, China. [7]College of Horticulture, Nanjing Agricultural University, Nanjing, China. [8]Boyce Thompson Institute, Cornell University, Ithaca, NY, USA. [9]US Department of Agriculture-Agricultural Research Service, Robert W. Holley Center for Agriculture and Health, Ithaca, NY, USA. [10]Biotechnology Research Institute, Chinese Academy of Agricultural Sciences, Beijing, China. [11]These authors contributed equally: Ning Li and Qiang He. ✉e-mail: wanghuan@caas.cn; lihongbo_solab@163.com; yuqinghui@xaas.cn

**Table 1 | Assembly and annotation statistics of the 13 tomato genomes**

| Accession | Assembly size (Mb) | Percentage of anchoring (%) | Contig N50 (kb) | No. of predicted genes | Repeats (%) | BUSCO (%) |
|---|---|---|---|---|---|---|
| *S. lycopersicoides* (LA2951) | 1,200 | 92.23 | 579 | 32,295 | 71.81 | 90.1 |
| *S. habrochaites* (LA1777) | 960 | 86.07 | 546 | 32,386 | 69.05 | 92.0 |
| *S. pennellii* (LA716)[a] | 990 | 93.62 | 46 | 44,965 | 64.82 | 96.3 |
| *S. chilense* (LA1969) | 917 | 88.11 | 425 | 34,375 | 73.70 | 94.9 |
| *S. peruvianum* (LA0446) | 867 | 91.90 | 678 | 31,877 | 73.83 | 94.9 |
| *S. corneliomulleri* (LA1331) | 877 | 88.60 | 449 | 31,692 | 74.49 | 94.2 |
| *S. neorickii* (LA0247) | 778 | 94.07 | 2,079 | 32,831 | 72.74 | 93.1 |
| *S. chmielewskii* (LA1028) | 770 | 95.44 | 2,002 | 31,613 | 72.26 | 94.4 |
| *S. pimpinellifolium* (LA1547) | 803 | 94.78 | 3,691 | 33,427 | 72.77 | 93.1 |
| *S. galapagense* (LA0436) | 802 | 99.56 | 15,538 | 32,773 | 71.45 | 96.7 |
| *S. lycopersicum* var. *cerasiforme* (LA1464) | 778 | 94.68 | 2,513 | 32,941 | 73.50 | 96.7 |
| *S. lycopersicum* var. *lycopersicum* (M82) | 881 | 76.83 | 600 | 31,773 | 64.31 | 93.5 |
| *S. lycopersicum* var. *lycopersicum* (Heinz 1706)[a] | 828 | 97.48 | 6,008 | 35,768 | 63.46 | 96.4 |

[a]Genomes reported in previous studies.

By contrast, wild tomatoes in *Solanum* section *Lycopersicon*, which have adapted to various ecological environments in western South America including the Galapagos islands, from offshore to 3,600 m above sea level[3], exhibit broad genetic and phenotypic diversity[4–6]. These wild species represent a rich source of allelic variation and harbor genes underlying biotic and abiotic stress tolerance, as well as consumer-preferred traits such as high levels of soluble solid content, lycopene and flavor compounds[2,7]. Hence, effective harnessing of natural diversity from these wild germplasms is essential to facilitate tomato genetic improvement.

The availability of the tomato reference genome (*S. lycopersicum* var. *lycopersicum* cv. Heinz 1706)[8,9] has enabled comprehensive characterization of genetic diversity in terms of SNPs and small insertion/deletions (indels) by resequencing numerous accessions, revealing the domestication and wild introgression history of tomato[10,11]. Despite this, increasing numbers of studies have indicated that large structural variants (SVs), such as presence/absence variants and copy number variants (CNVs), also have vital roles in plant adaptive evolution and functional diversity[12,13]. However, conventional strategies based on a single linear reference genome can only capture a portion of genetic diversity, resulting in strong reference biases, and accurate detection of SVs is still challenging using merely short-read resequencing approaches.

To overcome these limitations, pangenomics, as applied in human and many plant species, has emerged as a promising approach to capture the nearly full spectrum of genetic diversity of crops and their wild relatives[14,15]. Recent genomic advances in tomato include a pangenome of 725 tomato accessions constructed using short reads[13], a pan-SV map built from Oxford Nanopore long reads of 100 diverse tomato lines[12] and a graph pangenome integrating variant information from 838 tomato genomes[16]. These studies suggest that SVs contribute to phenotypic variance and can be powerful when utilized in genome-wide association studies (GWAS). However, most of the accessions sampled in the studies were domesticated tomatoes and their closely related progenitor species *Solanum pimpinellifolium*. Currently, genome assemblies for five wild tomato species, *Solanum habrochaites*[11,17], *Solanum pennellii*[11,18,19], *Solanum galapagense*[17], *S. pimpinellifolium*[20,21] and *Solanum lycopersicoides*[22], are available, with different assembly approaches and qualities, which impedes the characterization

and utilization of genetic variants in tomato wild relatives. Recently, it was highlighted that a super-pangenome that includes genomic information of many diverse species, especially wild relatives within a genus, could expedite crop improvement[23]. Therefore, it is necessary to assemble additional reference genomes for tomato wild relatives to accelerate biological studies and genetic improvement in tomato. In this study, we construct a section-wide super-pangenome by de novo assembling 11 chromosome-level genomes from ten tomato species, representing major clades of tomato wild relatives and their cultivated counterparts in *Lycopersicon*. Comparative analyses reveal the panorama of genomic content, evolutionary history and structural variation across tomato species, empowering the discovery of a wild tomato gene that has the potential to increase yield in modern cultivated tomatoes. These results will provide insight for the construction and exploitation of super-pangenomes in other crop species.

## Results

### Eleven wild and cultivated tomato reference genomes

To represent the diversity of wild and cultivated tomato species, we selected nine wild tomatoes (eight species from *Solanum* section *Lycopersicon*: *S. habrochaites*, *Solanum chilense*, *Solanum peruvianum*, *Solanum corneliomulleri*, *Solanum neorickii*, *Solanum chmielewskii*, *S. pimpinellifolium* and *S. galapagense*; and one from *Solanum* section *Lycopersicoides*: *S. lycopersicoides*) and two diverse domesticated tomatoes (*S. lycopersicum* var. *cerasiforme* and *S. lycopersicum* var. *lycopersicum* cv. M82; Table 1). We assembled a high-quality chromosome-scale reference genome of wild tomato *S. galapagense* 'LA0436', using a hybrid assembly approach integrating Pacific Biosciences (PacBio) sequencing, optical genome mapping (Bionano Genomics) and high-throughput chromosome conformation capture (Hi-C; Supplementary Note and Supplementary Tables 1–5). The 802-Mb final assembly had a contig N50 length of 15.5 Mb, and more than 99.5% of sequences in the final assembly were anchored to the 12 chromosomes, higher than the corresponding percentages for the three existing reference genomes 'LA2093' (99.0%), 'Heinz 1706' (97.5%) and 'LA716' (93.6%) (Table 1). The ten other tomato genomes were also assembled at chromosome level using the above-mentioned strategy, except that Bionano data were not generated. These

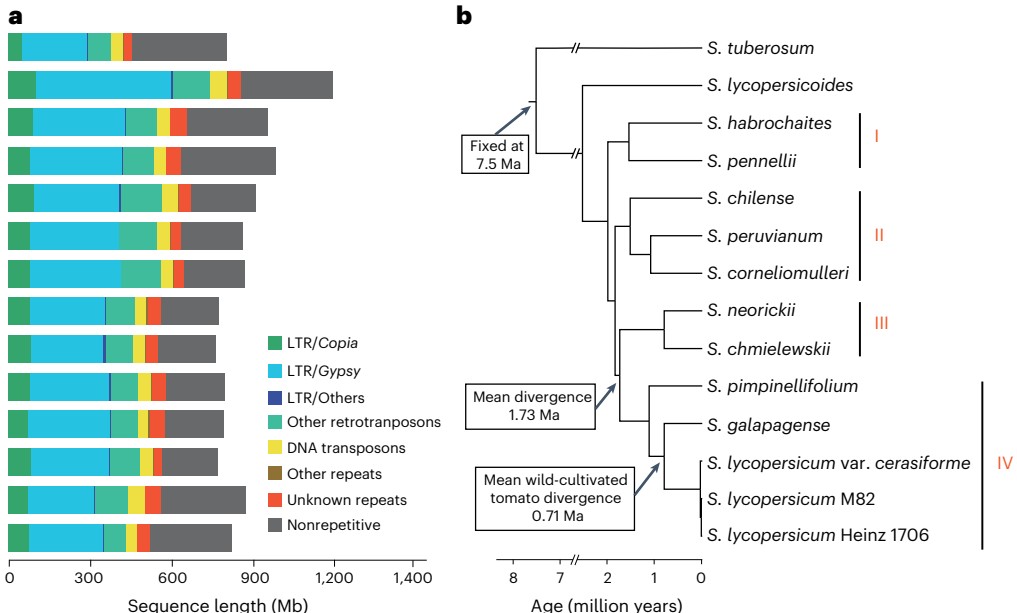

**Fig. 1 | Phylogenetic relationships and genomic components of wild and domesticated tomatoes. a**, TE content in genomes of potato and the 13 wild and cultivated tomatoes. The order of species is corresponding to the phylogeny shown in **b. b**, Species phylogeny of ten species (13 genomes) from *Solanum* sect. *Lycopersicon* and *Solanum* sect. *Lycopersicoides* using *S. tuberosum* as the outgroup. The 12 *Lycopersicon* genomes are clustered into four clades.

genomes had monoploid assembly lengths ranging between 770.0 Mb (*S. chmielewskii*) and 1.2 Gb (*S. lycopersicoides*), close to their predicted genome sizes (Table 1 and Supplementary Tables 6 and 7). More than 99% of Illumina short reads and 95.7% of ESTs could be mapped to the 11 tomato genome assemblies, and 94.0% of embryophyte Benchmarking Universal Single-Copy Orthologs (BUSCO)[24] were captured in these assemblies, indicative of their high completeness (Supplementary Tables 8–10).

We combined ab initio prediction, homology search and transcriptome mapping approaches for protein-coding gene prediction (Methods), resulting in gene numbers ranging from 31,613 (*S. chmielewskii*) to 34,375 (*S. chilense*), similar to that of Heinz 1706 (35,768) but fewer than that of LA716 (44,965) (Table 1). A total of 81.7% to 89.5% of exons of the predicted genes were supported by transcript data, suggesting the high quality of gene predictions. All assembled genome sequences and annotations are publicly accessible through a web-based database (http://caastomato.biocloud.net).

Eukaryotic genomes are rich in transposable elements (TEs), which shape genome evolution through expansions, eliminations and transpositions[25]. The TE contents of the 11 tomato genomes ranged from 64.3% to 74.5%, with long terminal repeat retrotransposons (LTR-RTs) representing the most abundant class of TE (Fig. 1a). A higher abundance of *Gypsy* LTR-RTs was found in *S. lycopersicoides*, which possibly contributed to it having the largest assembled genome size (1.2 Gb) among the tomato species[22] (Fig. 1a). To trace the evolutionary history of the expanded TEs in *S. lycopersicoides*, we estimated insertion times of 162,216 intact LTR-RTs and detected a lineage-specific burst of *Gypsy* LTR-RTs occurring c. 2 million years ago (Ma) in *S. lycopersicoides*, after its divergence from potato, probably leading to its large extant genome (Supplementary Fig. 3). Notably, we observed recent amplification of *Gypsy* and *Copia* LTR-RTs in four wild tomato species (*S. lycopersicoides*, *S. corneliomulleri*, *S. peruvianum* and *S. chilense*; Supplementary Fig. 3), implying that these wild species may have increasing degrees of genomic diversity and environmental adaptability compared with cultivated tomatoes. These results provide insight into the role of TEs in genome evolution of the *Solanum* genus.

## Phylogeny of *Lycopersicon* and neighboring species
Reconstructing the phylogeny of *Lycopersicon* species has been problematic owing to the conflict between gene trees and morphological trees, especially for the wild tomato clade[3]. The phylogenetic relationship between *S. pennellii* and other tomatoes remains unresolved[6], owing largely to limited available genomic data, despite *S. pennellii* being considered to be a unique group based on morphological classification. Using 9,343 single-copy orthologous genes, we inferred the phylogeny of ten wild and three domesticated tomatoes, using potato (*Solanum tuberosum*) as an outgroup; the results indicated that section *Lycopersicoides* (including *S. lycopersicoides*) was sister to section *Lycopersicon* (Fig. 1b), consistent with previous research[3]. Based on the phylogeny, we resolved the polytomy issue in *Lycopersicon* and unambiguously classified *Lycopersicon* species into four main clades. Clade I encompassed two species, *S. pennellii* and *S. habrochaites*, which diverged from the common ancestor of the other wild and cultivated tomatoes (except *S. lycopersicoides*) c. 1.97 Ma. Clade IV, which comprised domesticated tomatoes and two closely related wild species (*S. galapagense* and *S. pimpinellifolium*), divided from the ancestor of clade III (*S. neorickii* and *S. chmielewskii*) approximately 1.73 Ma. Similar to a recent study of *Oryza* genus evolution[26], a few conflicts were observed between the phylogeny constructed using genes from one chromosome and that built using whole-genome genes (Fig. 1b and Supplementary Fig. 4). For example, within *Lycopersicon*, phylogenetic analyses using genes from chromosomes 1, 2, 9 and 11 showed that *S. pennellii* was sister to other wild and cultivated tomato species, rather than clustering into a monophyletic group with *S. habrochaites* as inferred from the genome-wide phylogeny (Supplementary Fig. 4), suggesting possible incomplete lineage sorting and/or hybridization events. These results enhance our understanding of the evolutionary history within *Solanum* section *Lycopersicon*.

## Super-pangenome of tomato
Although pangenomes for cultivated tomato and its close wild relatives have been reported[13], the gene pool of *Lycopersicon*, which contains wild and cultivated tomato species, remains largely inaccessible. Here, we extended the tomato pangenome that integrates genomes from

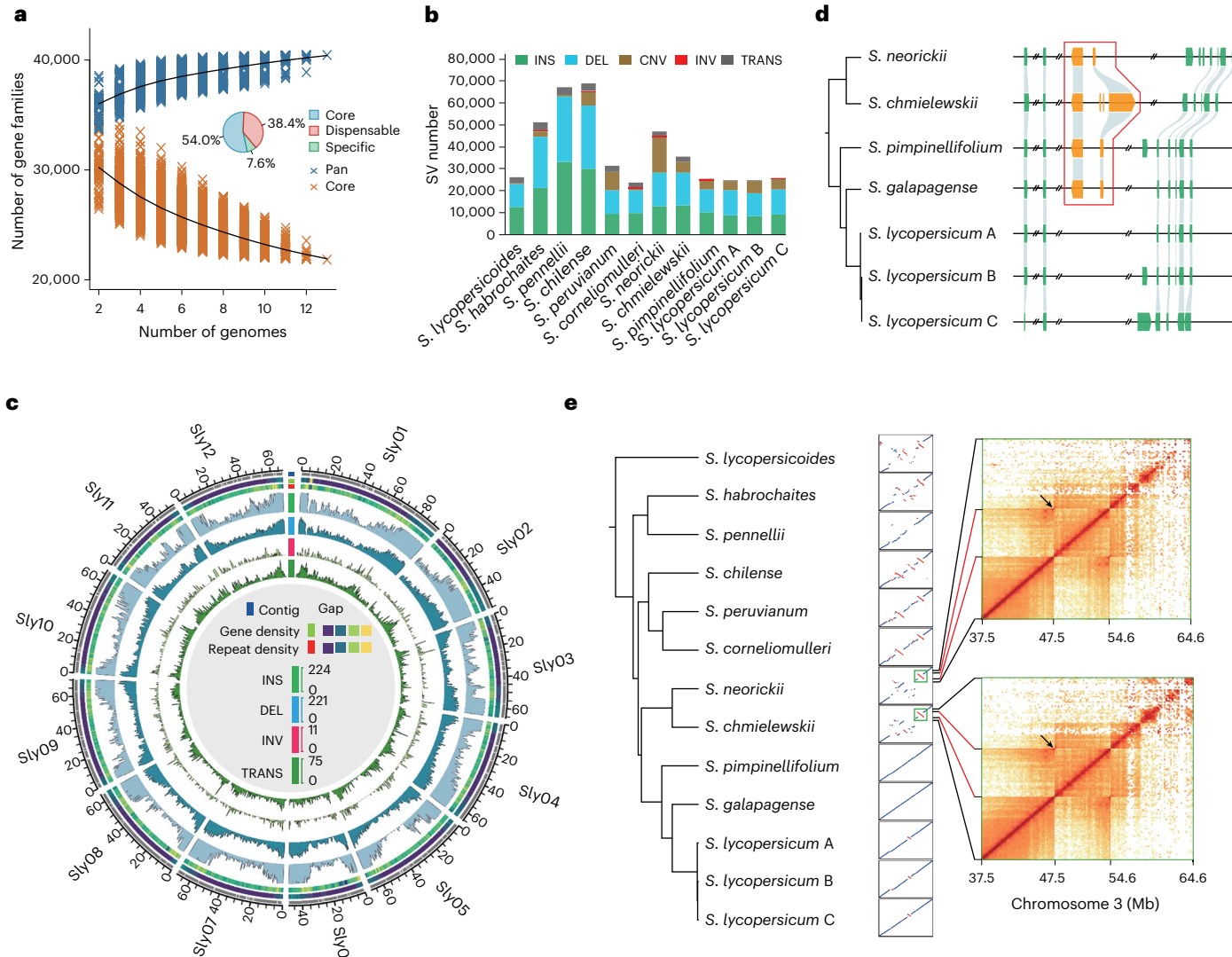

**Fig. 2 | Super-pangenome and the landscape of structural variation among wild and cultivated tomatoes. a**, Modeling of pangenome and core-genome sizes when incorporating additional genomes into clustering and composition of the tomato super-pangenome (pie chart). **b**, Number of different types of structural variants within each genome compared with the *S. galapagense* reference genome. **c**, Distribution of structural variants from the 12 tomato genomes across the 12 chromosomes. **d**, A wild-specific genomic fragment on chromosome 1. An 8-kb sequence was present in genomes of all nine wild tomatoes but absent from the three domesticated tomatoes. The 8-kb wild-specific region harboring two genes is outlined in red. **e**, Dot plots display the alignments of chromosome 3 between the 12 tomato genomes and the *S. galapagense* genome. A clade IV-specific inversion on chromosome 3 from 47.5 Mb to 54.6 Mb is shown, as evidenced by abnormally strong interactions around the inversion breakpoints in Hi-C heat maps. In **b**, **d** and **e**, *S. lycopersicum* A, *S. lycopersicum* var. *cerasiforme*. *S. lycopersicum* B, *S. lycopersicum* var. *lycopersicum* cv. M82. *S. lycopersicum* C, *S. lycopersicum* var. *lycopersicum* cv. Heinz 1706. DEL, deletion; INS, insertion; INV, inversion; TRANS, translocation.

three *Solanum* species[13] to a super-pangenome covering 11 species in the *Solanum* genus. We defined 40,457 pangene families by clustering protein-coding genes of the 11 chromosome-scale genomes assembled herein and two previously released genomes[8,18]; this number of gene families was higher than that of the *Oryza* genus[26] but lower than that of soybean[27]. The number of gene families increased rapidly when including more genomes, suggesting that the 13 genomes are diverse and that a single reference genome cannot capture the full genetic diversity in tomato (Fig. 2a). Only 54.0% of gene families were conserved among the 13 tomato genomes (core gene families), and the number of core genes (23,839) was lower than that of the previously reported pangenome of 519 cultivated tomatoes and 67 closely related wild tomato accessions belonging to *S. pimpinellifolium* and *S. galapagense* (29,938)[13], owing largely to the higher level of divergence among the wild tomato species

used in this study. The dispensable gene families (present in two to 12 accessions) occupied 38.4% of gene families, and 7.6% of pangene families were categorized as accession-specific.

Gene ontology (GO) enrichment analysis showed that core genes were enriched for biological processes including carboxylic acid, lipid or organic substance metabolic process, RNA modification or processing and amide transport, consistent with the results of a previous study[13] (Supplementary Tables 11 and 12), whereas the dispensable genes were enriched for terpenoid biosynthesis, telomere maintenance, mitochondrial electron transport and photosynthesis (Supplementary Fig. 6). Expression levels of core genes were significantly higher than those of dispensable genes at different fruit ripening stages ($P < 2.2 \times 10^{-16}$, Wilcoxon rank-sum test; Supplementary Fig. 7). We found that 3,441 out of the 4,874 nonreference genes reported from the

previous tomato pangenome[13] were captured in our super-pangenome (Supplementary Table 13), and we also identified 9,320 nonredundant genes absent from the reported tomato pangenome[13] (Supplementary Note and Supplementary Table 14), indicating the rich diversity of the 13 wild and domesticated tomatoes. This super-pangenome dataset lays a foundation for exploration and exploiting of genes or alleles in wild tomato species.

### Extensive variation among wild and cultivated tomatoes

Despite efforts to characterize genetic variants among cultivated tomatoes and their proposed progenitor species *S. pimpinellifolium*[12,13,16], the genetic diversity among distantly related wild tomato species, for example, *S. peruvianum, S. habrochaites* and *S. chilense*, remains poorly explored. We identified 2.0–8.1 million SNPs and 0.6–1.5 million small indels (≤50 base pairs (bp) in size) in the 12 tomato genomes, relative to the reference *S. galapagense* genome. The total number of SNPs and small indels (42.4 M) was much higher than that of each accession (Supplementary Tables 15 and 16), suggesting a diverse nature among the 12 wild and cultivated tomato accessions (Supplementary Note). Leveraging genome alignments, we identified 103,333 insertions, 119,794 deletions, 41,960 CNVs, 23,516 translocations and 1,320 inversions (<1 Mb in length) in the 12 tomato accessions compared with the *S. galapagense* genome (Supplementary Tables 17 and 18). Species in clade II (*S. chilense, S. peruvianum* and *S. corneliomulleri*) contained markedly varied numbers of SVs (Fig. 2b), possibly associated with the recent proliferation of LTR-RTs in those genomes (Supplementary Fig. 3). The majority of insertions, deletions and CNVs were shorter than 2 kb, 2 kb and 8 kb, respectively, and most of the translocations had lengths shorter than 20 kb, whereas some inversions were longer than 300 kb (Supplementary Fig. 17). We found that insertions and deletions were more likely to be found at both ends of the chromosomes, consistent with previous studies[12,20], whereas inversions and translocations were randomly distributed along the 12 chromosomes (Fig. 2c). SVs were more likely to occur at repeat regions than nonrepeat genomic regions (Student's *t* test, $P = 1.03 \times 10^{-4}$). We further identified 5,186 large indels (>50 bp) fixed either in all wild or all domesticated tomato genomes investigated in this study, some of which led to insertions of protein-coding genes present only in certain wild tomato genomes (Supplementary Table 19 and Fig. 2d). Further functional characterization of these variants may enable a better understanding of the genetic basis of phenotypic divergence between domesticated tomatoes and their wild relatives.

Previous studies have identified several SVs responsible for phenotypic variation, including a 1.4-kb deletion in the *CSR* gene resulting in increased fruit weight[28], a 7.1-kb deletion in the *LNK2* locus responsible for a light-conditional clock deceleration[29], an 85-bp deletion in the promoter of *ENO* that regulates floral meristem activity[30] and a CNV affecting *NSGT* associated with biosynthesis of a fruit flavor volatile guaiacol[12]. These SVs were all accurately detected in this study (Supplementary Figs. 18–21), indicating the broad diversity of our collection. Two different alleles (4,724 bp and 4,151 bp) have been identified at 149 bp upstream of *TomLoxC*, a gene encoding a 13-lipoxygenase; the 4,151-bp allele was reported to contribute to desirable fruit flavor and is rare in cultivated tomatoes[13]. We found that *S. pennellii, S. habrochaites, S. chilense* and *S. neorickii* carried the 4,151-bp allele upstream of *TomLoxC* (Supplementary Fig. 22), suggesting that these wild species have the potential to improve fruit flavor in cultivated tomato by backcrossing. The extensive variation among wild and cultivated tomato species presented herein provides access for further harnessing of the genetic diversity of distantly related wild tomato species in genomic-based breeding.

### Hidden genetic diversity of tomato wild species

Large inversions have been reported to suppress recombination by reducing crossing-over[31,32], resulting in severe linkage drag when conducting backcross breeding. To overcome this, it is necessary to choose donor lines without inverted segments harboring targeted genes. However, a holistic view of genome-wide inversions is not available, owing to the lack of chromosome-scale wild tomato genomes. Based on the 11 high-quality tomato genomes, we identified 12 (*S. lycopersicum* var. *lycopersicum* cv. Heinz 1706) to 42 (*S. chmielewskii*) megabase-scale inversions compared with the *S. galapagense* genome (Supplementary Table 20). Notably, a 7.1-Mb inversion on chromosome 3, carrying 55 genes, was present in all clade IV tomato accessions compared with other wild species (except *S. pennellii*) and was supported by clear chromatin interactions around the breakpoints when Hi-C reads of *S. neorickii* and *S. chmielewskii* were mapped to the *S. galapagense* genome (Fig. 2e). This inversion might occur after the divergence between species from clade IV and other clades. Given that *S. pennellii* does not carry this inversion within this region, this wild tomato species would be an ideal donor parent to introduce possibly favored genes within this 7.1-Mb segments into elite cultivars by backcrossing.

Previous research reported a tomato pan-SV map, which was built by long-read sequencing of 100 cultivated and closely related wild tomato accessions[12]. Compared with this pan-SV map, 180,314 out of the 224,447 SVs were exclusively identified in this study, of which 4,124 (2.3%) were localized within coding regions (CDS) of 3,515 genes (Supplementary Note and Supplementary Table 21), suggesting that the majority of SVs found in this study were captured owing to the inclusion of distantly related wild tomato species. Integrating our identified SVs with the pan-SV dataset generated 153,873 insertions, 203,364 deletions, 2,952 inversions and 45,987 duplications in 112 tomato accessions (12 in this study and 100 in the pan-SV map), allowing us to investigate the divergence of SVs during tomato evolution. We divided these 112 accessions into four groups: wild (19 non-*S. pimpinellifolium* wild accessions), SP (22 *S. pimpinellifolium* accessions), SLC (24 *S. lycopersicum* var. *cerasiforme* accessions) and SLL (47 big-fruited *S. lycopersicum* var. *lycopersicum* accessions; Supplementary Fig. 24a). The vast majority of SVs displayed relatively low frequencies (<0.25) in all four groups, and accessions from the wild group contained a higher proportion of SVs with presence frequency between 0 and 0.25 (Supplementary Fig. 24b). We observed that 8,094 SVs exhibited significant frequency changes between the wild and cultivated (SLL and SLC) groups (Fisher's exact test, false discovery rate (FDR) < 0.01; Supplementary Fig. 24c), affecting upstream regions and exons of 2,585 genes. Functional analyses indicated that these genes were mainly enriched for biological processes such as meristem development and ammonium transport (Supplementary Fig. 24d). We further identified 388 highly divergent SVs between wild and cultivated tomatoes, which disrupted CDS of 328 genes by causing frameshift, loss of exons or in-frame insertions (Supplementary Table 22). These results suggest that SVs in these distantly related wild tomatoes have undergone distinct evolutionary trajectories compared with cultivated tomatoes and their progenitors. Our analyses also provide a candidate dataset for further characterizing genes underlying phenotypes with great divergence between wild and cultivated tomatoes.

### A wild tomato cytochrome P450 gene that increases yield

A major goal of tomato breeding is to increase yield by developing varieties with larger fruit size and/or more effective shoot branches. Regulation of shoot architecture is thus of great interest to the tomato research community[33]. Wild tomato species usually display a markedly greater number of lateral fruit-bearing branches than their domesticated counterpart; however, whether we can introduce this trait into cultivated tomatoes, especially modern processing tomato varieties, remains elusive. Among the 388 highly divergent SVs between wild and cultivated tomatoes that greatly affected gene CDS, a 244-bp deletion, showing the second most significant frequency change (FDR = $1.43 \times 10^{-8}$; Supplementary Fig. 24c), was present in the first exon of *Sgal12g015720* (Fig. 3a,b). This gene encodes a protein belonging to the cytochrome

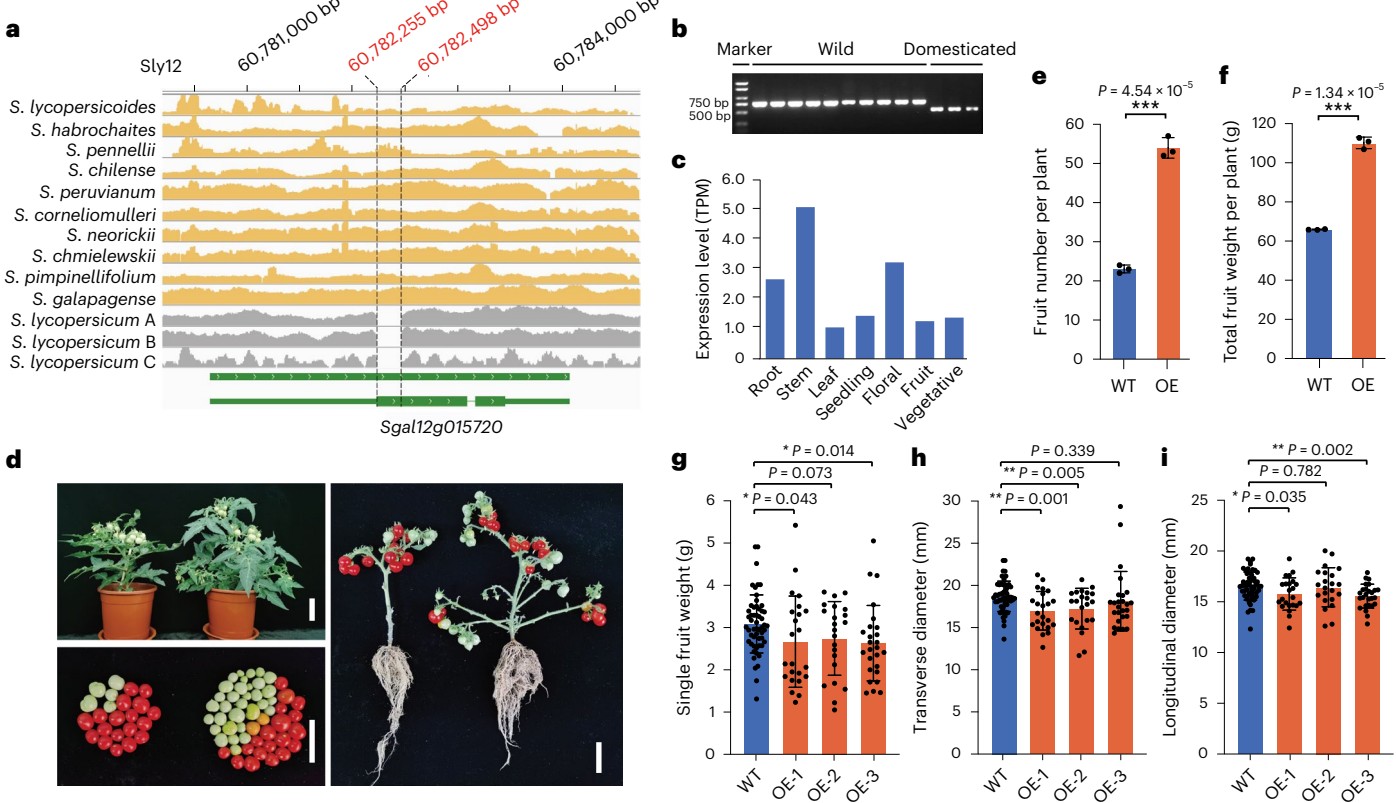

**Fig. 3 | Characterization of a wild tomato cytochrome P450 gene, Sgal12g015720. a**, A 244-bp deletion in the first exon of *Sgal12g015720* in the three domesticated tomatoes. Genome coverages when mapping Illumina reads against the *S. galapagense* reference genome are illustrated by yellow (ten wild species) and gray (three cultivated accessions) histograms. Green lines, 5′ and 3′ UTRs; bold green lines with white arrows inside, exons; light green lines, introns; *S. lycopersicum* A, *S. lycopersicum* var. *cerasiforme*; *S. lycopersicum* B, *S. lycopersicum* var. *lycopersicum* cv. M82; *S. lycopersicum* C, *S. lycopersicum* var. *lycopersicum* cv. Heinz 1706. **b**, PCR validation of the 244-bp deletion in ten wild and three domesticated tomatoes. Three experiments were independently conducted with similar results. **c**, Expression levels (transcripts per million

(TPM)) of *Sgal12g015720* in different tissues of wild tomato *S. pennellii*. **d**, Comparison of phenotypes of the WT Micro-Tom (left panel) plant and the $T_2$ generation of the *Sgal12g015720*-OE transgenic plant (right panel). Scale bar, 5 cm. **e–i**, Fruit number per plants (**e**), total fruit weight per plant (**f**), single fruit weight for red fruits (**g**), transverse diameter for red fruits (**h**) and longitudinal diameter for red fruits (**i**) in WT and $T_2$ transgenic plants. For **e** and **f**, three independent WT and OE plants are used. In **g–i**, the number of fruit samples for WT is 55 and numbers of fruits for OE-1, OE-2 and OE-3 are 23, 22 and 26, respectively. Data are presented as mean ± s.d.; \*\*\**P* < 0.001; \*\**P* < 0.01; \**P* < 0.05 in two-tailed Student's *t* test.

P450 superfamily, which has been reported to play important parts in plant growth, development and secondary metabolite biosynthesis[34]. The 244-bp deletion was found in 22.22% of the 19 wild accessions and 100% of cultivated tomatoes, which represented the derived state, as this deletion was absent from all the nine wild tomato species used in this study (Fig. 3a,b and Supplementary Fig. 25). *Sgal12g015720* was expressed at the highest level in stems of the wild tomato *S. pennellii* (Fig. 3c), whereas its expression in two cultivated tomatoes could barely be detected (Supplementary Fig. 26). These results suggest that the 244-bp deletion event may have occurred during tomato domestication, which might lead to pseudogenization of *Sgal12g015720* in cultivated tomato.

To investigate functions of this gene and its potential value for tomato breeding, we generated *Sgal12g015720*-overexpression (OE) transgenic lines under the background of a tomato cultivar 'Micro-Tom' (Supplementary Fig. 27). Compared with the wild-type (WT) plants, the transgenic lines possessed a greater number of lateral branches, resulting in a greater than twofold increase in total fruit number, whereas only slight reductions in single fruit weight, transverse diameter and longitudinal diameter were observed (red fruits; Fig. 3d–i). To further validate the function of *Sgal12g015720*, we screened previously reported introgression lines (ILs)[35] generated using wild tomato *S. pennellii*

(LA716, donor parent) and cultivated accession M82 (recurrent parent). As expected, two ILs, IL12-2 and IL12-3, carrying a homozygous introgressed segment that harbors an *Sgal12g015720* ortholog from the wild tomato donor, generated markedly more lateral branches and fruits compared with the recurrent parent M82 (Supplementary Fig. 28). Therefore, this gene represents a promising target for regulation of plant architecture as well as increasing yield or biomass in tomato breeding. These analyses also present an example of how the super-pangenome could facilitate tomato biological studies and breeding.

### Graph-based genome enables SV-based GWAS in tomato

Numerous studies have suggested that SVs are causative variants responsible for agronomically important traits[12,13,36,37]. However, population-scale SV genotyping is challenging in plants, impeding the exploitation of SVs in identifying genotype–phenotype associations. Here, we constructed a tomato graph-based genome by integrating the linear reference genome sequence of *S. galapagense* and the 360,189 SVs identified from the 12 tomato genomes and the 100 previously reported tomato genomes[12]. Graph-based genomes are capable of storing both reference and alternative allele sequences while retaining the coordinate systems of the linear reference genome, which facilitates

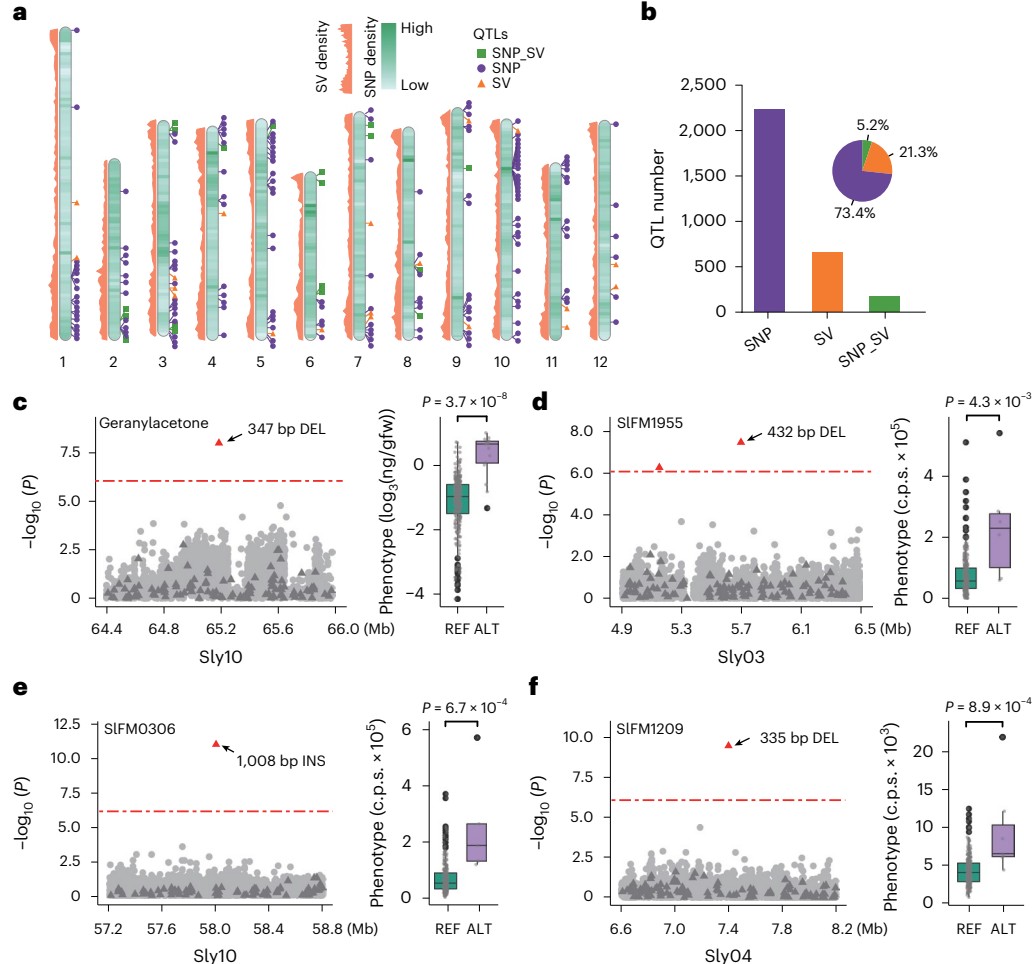

**Fig. 4 | SV-based GWAS identify additional association signals for tomato fruit flavor. a**, Density of SNPs and SVs used in GWAS and genome-wide distribution of quantitative trait loci (QTLs) with top 200 PVE. **b**, Number of QTLs detected by different categories of markers. SNP, QTLs that could only be identified by SNPs; SV, QTLs that could only be identified by SVs; SNP_SV, QTLs detected by both SNPs and SVs. **c–f**, Local Manhattan plots for geranylacetone content (**c**), SlFM1955 (kaempferol-sinapylglucosyl-xylosylrhamnoside (**d**), SlFM0306 (2'-deoxyadenosine monohydrate (**e**) and SlFM1209 (tricin 7-*O*-hexoside) (**f**) (left panel), and corresponding box plots in accessions carrying distinct alleles (right panel). In Manhattan plots, triangles denote SVs and points illustrate SNPs. Genome-wide threshold for GWAS ($7.58 \times 10^{-7}$) is marked using red dashed lines. In box plots, the 25% and 75% quartiles are shown as lower and upper edges of boxes, respectively, and central lines denote the median. The whiskers extend to 1.5 times the interquartile range. Data beyond the end of the whiskers are displayed as black dots. *P*-values were computed from two-tailed Student's *t* test. ng/gfw, nanograms per gram of fresh weight; c.p.s., counts per second; REF, accessions with homozygous reference (*S. galapagense*) type of allele; ALT, accessions possessing homozygous alternative allele. Numbers of samples for REF in box plots in **c**, **d**, **e** and **f** are 305, 288, 290 and 280, respectively. Numbers of samples for ALT in box plots in **c**, **d**, **e** and **f** are 16, 5, 5 and 7, respectively.

mapping of short reads from SV regions and thus SV genotyping[38,39]. We then genotyped these SVs in a tomato population comprising 321 accessions[2] and performed SV-based GWAS for 32 flavor-related compounds[2] and 362 fruit metabolites[40]. For comparison, we also called SNPs and indels from the 321 accessions and employed SNP-based GWAS.

Significantly associated signals were detected for 17 flavor volatiles and 249 fruit metabolites. Surprisingly, we observed that only 5.2% (161) of peaks (quantitative trait loci) overlapped (800-kb flanking region) between SV-based and SNP-based GWAS results, and 21.3% (658) could only be identified by SVs. The remaining 2,263 (73.4%) were exclusively detected by SNPs (Fig. 4a,b and Supplementary Table 23). Examples included a peak at 65.2 Mb on chromosome 10 that could only be detected using SVs, which was strongly associated with the content of geranylacetone ($P = 7.91 \times 10^{-9}$), one of the important tomato flavor volatiles contributing a leafy flavor to fruits (Fig. 4c and Supplementary Fig. 29). The leading SV was a 347-bp deletion, and the content of geranylacetone in tomato fruits significantly differed

between accessions carrying the reference allele and those carrying the alternative allele (Student's *t* test, $P = 3.7 \times 10^{-8}$, Fig. 4c). Similarly, we detected significantly associated SVs for the content of additional metabolites (Fig. 4d–f, Supplementary Figs. 30–32 and Supplementary Table 23). Tomato accessions carrying alleles of corresponding leading SVs showed significantly increased content of these metabolites (Fig. 4d–f). This identification of SVs exhibiting significant associations with important tomato fruit flavor compounds and metabolites will pave the way for further fine mapping and isolation of putative candidate genes. Our SV-based GWAS provide an important complement to the conventional SNP-based GWAS, which will be helpful to develop markers for breeding flavor-improved tomato cultivars.

## Discussion

Domestication of tomato has led to a substantial loss of genetic diversity in modern varieties due to the bottleneck and successive rounds of artificial selection; therefore, the rich diversity of wild tomato

species contains valuable breeding materials. However, the availability of only a few wild tomato genomes has hampered the exploration and utilization of alleles and gene repertoire in those wild species. The chromosome-scale reference genomes for nine wild tomato species presented here offer valuable resources for not only comparative genomics but also biological studies and molecular breeding in tomato. Notwithstanding, our dataset still lacks three wild tomato species in *Solanum* section *Lycopersicon* (*Solanum cheesmaniae*, *Solanum huaylasense* and *Solanum arcanum*). *S. cheesmaniae* is endemic to the Galápagos island with yellow to orange fruits[41], whereas *S. huaylasense* and *S. arcanum* are wild tomatoes segregated from *S. peruvianum*[42]. Development of their genome sequences and annotation will further enrich our understanding of the biodiversity and evolutionary trajectory within *Lycopersicon*.

Although pangenomes for numerous crops have been reported, most of them incorporated one or a few species[43]. Here, we constructed a super-pangenome by analyzing 11 distinct tomato species, representative of major wild and cultivated tomato clades. Coupling this with an existing dataset[12], we identified a wild tomato gene that could increase fruit yield by an average of 67.1% in OE transgenic lines (Fig. 3d–f). As both OE lines and ILs carrying this gene had higher numbers of fruit-bearing branches (Fig. 3d and Supplementary Fig. 27), we anticipate its use in modern processing tomatoes. According to tomato population resequencing data, this gene was predominantly found in wild tomato accessions (52% of *S. pimpinellifolium*, 80% of *S. cheesmaniae* and 100% of *S. galapagense*), in contrast to a mere 6% and 19% in cultivated tomato forms *S. lycopersicum* var. *lycopersicum* and *S. lycopersicum* var. *cerasiforme*, respectively (Supplementary Table 24). These results indicate that this gene, although potentially important, has not been widely utilized in tomato breeding programs. Backcrossing would be an ideal approach to introduce this gene into cultivated tomatoes from wild species. However, hybridization between wild and cultivated crops may lead to severe repression of genetic recombination, owing largely to the presence of large-scale genomic divergence, such as large inversions[10,31]. This may ultimately result in the introduction of exotic genomic fragments carrying unfavorable alleles that are hard to purge[40]. We did not observe chromosomal rearrangements between the genome of Heinz 1706 and those of eight out of the nine wild species surrounding this gene (Supplementary Table 25), suggesting that introgression of this gene by backcrossing, when the donor parent is properly selected, would be less likely to cause linkage drag.

To facilitate the utilization of genetic diversity from our super-pangenome, we constructed a graph-based genome reference for wild and cultivated tomatoes by integrating SV information for 112 tomatoes from 11 *Solanum* species into the linear reference sequence, offering a powerful platform for population-level SV genotyping. As previous research has suggested that SVs are more likely to be causal variants in tomato[16], further studies could use this graph-based genome to perform SV-based association analyses to identify additional signals responsible for agronomically important traits. However, the current graph-based tomato genome is only capable of storing certain types of SVs: insertions, deletions and inversions. Other SVs of relatively high complexity, such as inverted duplications and translocations, cannot yet be integrated. Furthermore, SVs with multiple alleles are not represented in the graph, as downstream analytic pipelines can only handle biallelic variants. It is possible that an insertion with distinct inserted fragments in various individuals contributes to different phenotypic outcome. We anticipate further implementation of relevant tools and algorithms that could tackle these issues. This research will accelerate comparative genomics and biological studies in tomato and shed light on the utilization of super-pangenomes in crop improvement.

## Online content

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

## Methods

### Plant materials

Briefly, eight wild species from section *Lycopersicon* (*S. galapagense*, *S. pimpinellifolium*, *S. chmielewskii*, *S. neorickii*, *S. corneliomulleri*, *S. peruvianum*, *S. chilense* and *S. habrochaites*), one wild species from section *Lycopersicoides* (*S. lycopersicoides*) and two domesticated tomatoes (*S. lycopersicum* var. *lycopersicum* cv. M82 and *S. lycopersicum* var. *cerasiforme*) were collected. All seedlings were planted in Anningqu field test station of Xinjiang Academy of Agricultural Sciences.

### De novo genome assembly

Methods for library construction and sequencing are provided in the Supplementary Note. Contig-level assemblies for the 11 representative accessions were conducted using a pipeline based on Canu (v.1.5)[16,44] with the following procedures: longer seed reads were selected with the settings corOutCoverage = 35; raw read overlapping was detected using a highly sensitive overlapper MHAP[45] (v.2.1.2, parameter corMhapSensitivity = normal), and error correction was performed using the Falcon[46] sense method (option correctedErrorRate = 0.025); error-corrected reads were trimmed of unsupported bases and hairpin adapters to reach their longest supporting range with default parameters, and the draft assemblies were then generated using the top 80% longest trimmed reads. Finally, to ensure base accuracy of assembly results from SMRT molecules, we further polished the consensus genome sequences based on Illumina paired-end reads using Pilon[47] (v.1.22) with parameter: -mindepth 10 –fix bases.

### Scaffolding using Bionano optical maps

For *S. galapagense*, we constructed Bionano optical maps. Young leaves were collected after two days of dark treatment. High-molecular-weight DNA was isolated and labeled with the restriction endonuclease Nb.BssSI, and labeled DNA was imaged with a Bionano Irys system. Molecules with lengths >150 kb, label SNR ≥3.0 and average molecule intensity <0.6 were retained for scaffolding. These molecules were de novo assembled into genome maps using IrysSolve v.3.5_12162019 (https://bionanogenomics.com/support/software-downloads/). Pairwise comparison was first performed with RefAligner (https://bionanogenomics.com/support/software-downloads/) to identify overlaps among these molecules, and consensus maps were constructed. All molecules were then mapped back to the consensus maps and recursively refined and extended.

The Bionano IrysSolve module 'HybridScaffold' was used to perform hybrid assembly between the assembled contigs and genome maps. Assembled contigs were first converted into cmap format and then aligned to the contig cmaps with RefAligner, followed by label rescaling. The rescaled Bionano cmaps were aligned again to the contig cmaps, and sequences were split at the conflict points. Finally, scaffolds were built according to the alignment information. To improve the contiguity of assembly results, PBJelly[48] (v.15.8.24) was used to fill gaps using the error-corrected PacBio reads.

### Pseudomolecule construction

The Hi-C data were mapped to the assemblies using BWA[49] (v.0.7.10-r789) with default parameters. Only uniquely aligned read pairs with mapping quality >20 were retained for further analysis. Duplicate removal, sorting and quality assessment were performed using HiC-Pro[50] (v.2.8.1) with default parameters. Only valid interaction pairs of Hi-C reads were fed into LACHESIS (v.1.0)[51] for chromosome-scale scaffold construction. Briefly, contigs or scaffolds for each tomato assembly were broken into fragments with a length of 200 kb and then clustered using valid interaction read pairs by LACHESIS with the following parameters: 'CLUSTER_MIN_RE_SITES = 22, CLUSTER_MAX_LINK_DENSITY = 2, CLUSTER_NONINFORMATIVE_RATIO = 2, ORDER_MIN_N_RES_IN_TRUN = 10, ORDER_MIN_N_RES_IN_SHREDS = 10'. We manually checked the Hi-C interaction heat maps to identify potential

genomic regions containing assembled haplotigs due to heterozygosity, which were then excluded from the assembly. The manual curation step was reperformed several times, until the chromatin interaction signals reflecting putative haplotigs were undetectable.

### Evaluation of genome assemblies

Completeness of the assembled tomato genomes was first assessed using BUSCO[24] (v.5.2.0) based on the embryophyta_odb9 database. We also assessed the mapping proportions of transcripts assembled with Trinity (v.2.8.5)[52] to corresponding genome assemblies using BLASTN (v.2.12.0+)[53] with minimum alignment length of 300 bp and sequence identity >95%. These assemblies were also evaluated by mapping the Illumina short reads using BWA (default parameters).

### Repeat sequence annotation

Both homology-based and de novo strategies were applied to identify repetitive sequences for all the tomato genomes. Four de novo prediction programs were applied: RepeatScout[54] (v.1.0.5), LTR-FINDER[55] (v.1.05), MITE-hunter (v.1.0)[56] and PILER-DF[57] (v.1.0). Results from these four programs were integrated into a repetitive sequence database, which was then merged with Repbase[58] (v.19.06) and classified into different categories by the PASTEClassifier.py script included in REPET[59] (v.2.5). Using this repeat database, repetitive sequences were identified by homolog searching using RepeatMasker[60] (v.4.0.5) with default parameters. We computed the genetic distance (*K*) between both ends of an intact LTR-RT using the distmat (default parameters) program in the EMBOSS package (v.6.6.0)[61], and the insertion time (*T*) of each intact LTR-RT was estimated using the formula $T = K/2\mu$, where $\mu$ is the base substitution rate of $1.3 \times 10^{-8}$ (ref. [62]).

### Gene prediction and functional annotation

De novo, homology-based and transcriptome-based strategies were used to predict protein-coding genes for all tomato genomes assembled in this study. Predicted proteins from four plant genomes (*Arabidopsis thaliana*, *Oryza sativa*, *S. lycopersicum* and *S. tuberosum*) were used to perform homology-based prediction with GeMoMa[63] (v.1.3.1). Regarding de novo prediction, three different programs were used: GENSCAN (http://hollywood.mit.edu/GENSCAN.html, v.1.0), AUGUSTUS[51] (v.2.4) and GlimmerHMM[64] (v.3.0.4). We used AUGUSTUS with parameters trained by unigenes, which were assembled from pooled transcriptome data. As for the third approach, transcriptome data generated from pooled tissues of leaves, stems and roots were assembled using HISAT2 (ref. [65]) (v.2.0.4) and StringTie[66] (v.1.2.3), and the assembled contigs were aligned to the genome assemblies using BLAT (v.36)[67] (identity ≥0.95, coverage ≥0.90). The assembled contigs were then filtered using PASA[68] (v.2.0.4). We also mapped pooled transcriptome data to the reference genome using TopHat (v.2.0.12)[69] and performed reference-guided assemblies with Cufflinks (v.2.2.1)[70]. Transdecoder[71] (v.2.0) was then used to infer the structures of gene models and transcripts assembled by Cufflinks. By giving weights for the three methods, all predicted gene structures were synthesized into consensus gene models using EVidenceModeler[72] (v.1.1.1). All gene models were annotated according to their best BLASTP[43] (v.2.2.31; *E*-value <1 × 10⁻⁵) hits in protein databases including KEGG[73], Swiss-Prot[74], TrEMBL[74] and nonredundant protein database NR[75]. Blast2GO (v.4.1.8)[76] was used to assign GO terms for each gene.

### Phylogenetic tree construction and divergence time estimation

We selected *S. tuberosum* as the outgroup to infer species phylogeny. Single-copy orthologous genes were identified using quota-alignment (v.1.0)[77] with parameters '–merge–format=raw–Dm 30–Nm 40'. A total of 9,343 orthologous groups were identified among the 14 genomes. Protein sequences of the 9,343 single-copy orthologous genes were aligned using MUSCLE[78] (v.3.8.31) and the alignments were

then concatenated. We constructed a phylogenetic tree using phyML (v.3.3.20190909)[79] with parameters '−model JTT -f e -v 0.576 -a 0.886−nclasses 4−search SPR -t e'. The divergence time was estimated using the MCMCtree program in the PAML package[80] (v.4.7b). Three calibration points (*S. tuberosum* versus *S. lycopersicum* var. *cerasiforme*: 7.0–8.0 Ma; *S. lycopersicoides* versus *S. lycopersicum* var. *cerasiforme*: 2.0–2.7 Ma; and *S. pimpinellifolium* versus *S. lycopersicum* var. *cerasiforme*: 1.0–1.5 Ma)[81] were used to constrain the divergence time.

## Analyses of the super-pangenome

To identify homologous relationships among the genomes of 11 tomatoes assembled in this study, *S. lycopersicum* var. *lycopersicum* cv. Heinz 1706 and *S. pennellii*, the longest transcript of each predicted gene in each genome was chosen as a representative. To handle unannotated genes, a common issue during gene prediction, we aligned coding sequences of all predicted genes to each of the 13 tomato genomes using GMAP (v.2015-06-12)[82]. If a gene showed more than 80% alignment coverage and identity, and no gene was predicted within the aligned regions, it was considered to be an unannotated gene in the corresponding genome and was not regarded as 'missing' in the further analysis. An all-against-all comparison was then performed using BLASTP[53] (*E*-value <$1 \times 10^{-5}$), followed by clustering using OrthoFinder (v.2.5.2)[83] with default parameters. Based on the clustering results, we extracted gene families that were shared among all samples; these were defined as core gene families. Genes that were absent from two or more samples were defined as dispensable gene families, whereas those only present in one individual were considered to be specific gene families. Clade-specific gene families were defined as those exclusively present in one of the four clades of wild and cultivated tomatoes. Enrichment analysis with respect to GO terms was performed using the 'topGO' R package (https://bioconductor.org/packages/topGO). Details of the methods used for comparison of the super-pangenome and the previously reported tomato pangenome are provided in the Supplementary Note.

## Identification of genetic variants

We performed pairwise genome alignments between each of the 12 genomes and the *S. galapagense* reference genome using the nucmer program in MUMmer[84] (v.4.0.0beta2) with default parameters. The resultant alignments were filtered to retain the one-to-one alignment blocks, and SNPs and indels (<50 bp in length) were identified by the show-snps program within MUMmer with parameters '-Clr -x 1 -T'. For identification of SVs, two sets of SV calling results were generated using SVMU (v.0.4-alpha)[85] and SyRI (v.1.2)[86], respectively, both using default parameters. For SV detection using SVMU, intergenomic alignments were performed using the nucmer program in MUMmer[84] (v.4.0.0beta2) with default parameters. The results were then parsed in SVMU to produce collinear blocks and insertions, deletions and CNVs. Insertions and deletions larger than 50 bp inside the syntenic alignment regions were also kept for further analysis. For SV calling using SyRI, minimap2 (v.2.21-r1071)[87] was used to generate pairwise genome alignments with parameters '-ax asm5−eqx'. The alignment results were subsequently passed to SyRI, and SVs consisting of insertions, deletions, inversions (<1 Mb in size) and translocations (>50 bp in length) were kept. SVs encompassing 'N' sequences were removed. SVs with ambitious alignment margins and/or poor synteny alignment surrounding the breakpoint were also filtered. We only kept CNVs from SVMU output by applying a filtering of length >50 bp and coverage of reference or coverage of query ≥2 or ≤0.5. Inversions that were >1 Mb were extracted from the results generated from SyRI, followed by a manual check. The identified SVs from each sample were then merged using SURVIVOR (v.1.0.6)[88] with the following parameters: '50 1 0 0 0 0'.

## Analyses of presence frequency of SVs in tomato populations

Details of integrating SVs reported in the previous study are provided in the Supplementary Note. The 12 tomato genomes used in this study and the 100 previously reported tomato accessions[12] were divided into four groups: wild (19 accessions from *S. galapagense*, *S. cheesmaniae*, *S. chmielewskii*, *S. neorickii*, *S. corneliomulleri*, *S. peruvianum*, *S. chilense*, *S. habrochaites* and *S. lycopersicoides*), SP (22 *S. pimpinellifolium* accessions), SLC (24 *S. lycopersicum* var. *cerasiforme* accessions) and SLL (47 big-fruited *S. lycopersicum* var. *lycopersicum* accessions). We computed presence frequencies of each SV in the four groups and compared those between the wild and cultivated (SLC and SLL) groups using Fisher's exact test. The resultant *P*-values were next adjusted using the p.adjust function in R (v.4.03), with the 'method = 'fdr'' parameter. SVs with FDR < 0.01 were regarded as highly divergent between wild and cultivated tomatoes, showing significantly altered presence frequencies between the two groups.

## Functional characterization of the candidate gene

To generate an overexpression construct, the full-length ORF sequence of the candidate gene *Sgal12g015720* was amplified from *S. galapagense* using specific primers (Supplementary Table 26) and cloned into the plant expression vector pCAMBIA1300 by seamless cloning. Micro-Tom (*S. lycopersicum* var. *lycopersicum*) was transformed with the overexpressing transgene using *Agrobacterium tumefaciens* (strain GV3101)-mediated cotyledon transformation.

## Quantitative real-time PCR

Total RNA was isolated from young fresh materials (roots, stems and leaves) of WT and transgenic tomato lines using the a Plant RNA Kit (catalog number DP432, Tiangen), and cDNA sequences were synthesized using 5X All-In-One Master Mix (with AccuRT Genomic DNA Removal Kit; catalog number G492, Applied Biological Materials Quidel) according to the manufacturer's instructions. Real-time quantitative PCR (rt-qPCR) was carried out using a LightCycler96 real-time PCR system. Detection of rt-qPCR product was performed by staining with ChamQ SYBR qPCR Master Mix (catalog number Q311-02/03, Vazyme Biotech Co.). Specific primers are listed in Supplementary Table 26. The relative amplification of the tomato *actin* gene was used for normalization. The amplification was performed using the following conditions: 95 °C for 2 min followed by 40 cycles of 95 °C for 5 s and 60 °C for 30 s. Three samples (biological replicates) of each treatment were duplicated (technical replicates) in the rt-qPCR experiment. The relative expression level of genes was quantified according to the $R = 2^{-\Delta\Delta Ct}$ mathematical model. The final value of relative quantification was described as the fold change of gene expression in the test sample compared with the internal control (*actin*).

## Graph-based tomato genome construction and SV genotyping

To integrate the linear reference genome and large-scale genomic variant information, we constructed a graph-based genome of tomato using vg (v.1.38.0)[38]. Reference sequences of *S. galapagense* and SVs in terms of insertions and deletions from the 12 tomato genomes (this study) and the 100 tomato genomes reported from a previous study[12] were built into a variation graph by the 'construct' subcommand in vg without removing any alternate alleles. The preliminary graph was indexed in XG and GBWT by using 'vg index' with the '-L' option to retain alternative allele paths. A GBWT index was then built using 'vg gbwt' with the parameter '-P'. Previously reported Illumina paired-end reads of 321 tomato accessions were subsequently mapped against the indexed graph, and alignments in GAM format were generated by vg giraffe[89]. We then excluded low-quality alignments with mapping quality less than 5 and base quality less than 5. Finally, a compressed coverage index was calculated using 'vg pack', and snarls were generated using 'vg snarls', both with default parameters. SV genotyping in the 321 tomato accessions were performed using 'vg call' (default parameters) by examining the state (including read pair and split read information) and coverage of mapped reads around the SV breakpoints. Genotyped SVs with fewer than two supporting reads were marked as 'missing'.

## Genome-wide association studies

We selected the 321 tomato accessions that have been resequenced[2,10] for GWAS. A total of 43,901,591 SNPs were identified using the GATK (v.4.1.4.1) pipeline[90] with the *S. galapagense* genome as the reference. Population structure was calculated by principal component analysis in PLINK (v.1.9.0b4.6)[91] using 437,028 SNPs showing less linkage disequilibrium, which was extracted using PLINK with parameters '−indep-pairwise 50 5 0.1 (windows, step, $r^2$)'. The first five principal components were used as cofactors for population structure correction.

A total of 32 tomato flavor-related metabolite traits reported previously[2] and contents of 362 annotated metabolites from tomato fruits reported previously[40] were analyzed using EMMAX (v.20120210)[92] with the default KN kinship, in which the selected principal components were used as cofactors. SNP-based and SV-based GWAS were performed using SNPs or SVs with minor allele frequency >0.01 and missing call rate <0.1. The genome-wide significance thresholds ($7.58 \times 10^{-7}$) were determined using a uniform threshold of $1/n$, where $n$ is the effective number of independent SNPs and SVs calculated using the Genetic type 1 Error Calculator (v.0.2)[93]. Phenotypic variation explained (PVE) was calculated by the formula PVE = $[2 \times (beta^2) \times MAF \times (1 − MAF)]/[2 \times (beta^2) \times MAF(1 \cdot MAF) + ((s.e. \times (beta))^2) \times 2 \times N \times MAF \times (1 − MAF)]$, where $N$ represents sample size, s.e. is the standard error of the effect number of genetic variants, *beta* is the effect number of genetic variants and MAF is the minor allele frequency of the target marker.

## Reporting summary

Further information on research design is available in the Nature Portfolio Reporting Summary linked to this article.

## Data availability

All assembled genome sequences and annotations are accessible through our database (http://caastomato.biocloud.net). Assemblies for the tomato genomes have also been deposited in the National Center for Biotechnology Information (NCBI) under BioProject accession number PRJNA809001. Raw PacBio, transcriptome and Hi-C sequencing reads have been deposited in the NCBI sequence read archive (https://www.ncbi.nlm.nih.gov/sra/) under BioProject accession number PRJNA756391. Tomato whole-genome sequencing data were downloaded from NCBI (BioProjects: PRJNA259308, PRJNA353161, PRJNA454805 and PRJEB5235). The RepBase database was downloaded from https://www.girinst.org/server/RepBase/index.php. Source data are provided with this paper.

## Code availability

Custom scripts and codes used in this study are available at GitHub (https://github.com/HongboDoll/TomatoSuperPanGenome) and Zenodo (https://doi.org/10.5281/zenodo.7396707)[94].

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

## Acknowledgements

We thank J. Zhang (Institute of Vegetables and Flowers, Chinese Academy of Agricultural Sciences) for critical discussion of the project and revision of the manuscript. We thank J. Liu's group (Institute of Crop Sciences, Chinese Academy of Agricultural Sciences), E. Xia and W. Tong (Anhui Agricultural University), Y. Fang (Agricultural Genomics Institute at Shenzhen, Chinese Academy of Agricultural Sciences) and H. Du (Hebei University) for their helpful technical support with genome assembly and project discussion. We thank the Tomato Genetics Resource Center of UC Davis for providing seeds for wild tomato accessions. We thank M. Liu (Biomarker Technologies) and C. Ji (Chinese Academy of Tropical Agricultural Sciences) for providing sequencing platforms and support with data analysis. This research was supported by grants from the National Natural Science Foundation of China (grant nos. 31860555, 32260763 and 31991180), Key Projects for crop traits formation and cutting-edge technologies in biological breeding (xjnkywdzc-2022001), Key Research and Development Task special project of Xinjiang (2022B02002), Special Incubation Project of Science & Technology Renovation of Xinjiang Academy of Agricultural Sciences (xjkcpy-2021001), China Agriculture Research System of MOF and MARA (CARS-23-G24), Guangdong Major Project of Basic and Applied Basic Research (2021B0301030004), the National Key Research and Development Program of China (2019YFA0906200 and 2021YFF1000100), Shenzhen Science and Technology Program (grant no. KQTD2016113010482651), Special Funds for Science Technology Innovation and Industrial Development of Shenzhen Dapeng New District (grant no. RC201901-05), Shenzhen Outstanding Talents Training Fund and the US National Science Foundation (IOS-1855585).

## Author contributions

Q.Y., H. Li and H.W. conceived and designed the research. Q.Y. and S.H. managed the project. N.L., Q.H. and H. Li performed data analysis. J.Z. J.W., B.W., S.Y.H., Y.T., R.X., C.J., J.H., Z.L., F.J., J.G. and H. Lan collected the samples and performed experiments. Q.H. and N.L. wrote the draft manuscript. H. Li wrote the final manuscript. Q.Y., H. Li, Z.F., S.H., X.Z. and Y.Z. revised the manuscript. T.Y., S.Y., P.A. and K.M. supervised the field experiment.

## Competing interests

N.L., Q.H., B.W., J.W., Q.Y., T.Y. and P.A. have filed patent applications on technology related to the processes described in this article (Chinese patent application number CN202111231381.0; 'Application of the *BFNE* gene in improvement of plant architecture and biomass in tomato'). The other authors declare no competing interests.

## Additional information

**Correspondence and requests for materials** should be addressed to Huan Wang, Hongbo Li or Qinghui Yu.

# Reporting Summary

Nature Research wishes to improve the reproducibility of the work that we publish. This form provides structure for consistency and transparency in reporting. For further information on Nature Research policies, see our Editorial Policies and the Editorial Policy Checklist.

## Statistics

For all statistical analyses, confirm that the following items are present in the figure legend, table legend, main text, or Methods section.

| n/a | Confirmed | |
|---|---|---|
| ☐ | ☒ | The exact sample size (*n*) for each experimental group/condition, given as a discrete number and unit of measurement |
| ☐ | ☒ | A statement on whether measurements were taken from distinct samples or whether the same sample was measured repeatedly |
| ☐ | ☒ | The statistical test(s) used AND whether they are one- or two-sided<br>*Only common tests should be described solely by name; describe more complex techniques in the Methods section.* |
| ☒ | ☐ | A description of all covariates tested |
| ☒ | ☐ | A description of any assumptions or corrections, such as tests of normality and adjustment for multiple comparisons |
| ☐ | ☒ | A full description of the statistical parameters including central tendency (e.g. means) or other basic estimates (e.g. regression coefficient) AND variation (e.g. standard deviation) or associated estimates of uncertainty (e.g. confidence intervals) |
| ☐ | ☒ | For null hypothesis testing, the test statistic (e.g. *F*, *t*, *r*) with confidence intervals, effect sizes, degrees of freedom and *P* value noted<br>*Give P values as exact values whenever suitable.* |
| ☒ | ☐ | For Bayesian analysis, information on the choice of priors and Markov chain Monte Carlo settings |
| ☒ | ☐ | For hierarchical and complex designs, identification of the appropriate level for tests and full reporting of outcomes |
| ☒ | ☐ | Estimates of effect sizes (e.g. Cohen's *d*, Pearson's *r*), indicating how they were calculated |

*Our web collection on statistics for biologists contains articles on many of the points above.*

## Software and code

Policy information about availability of computer code

| Data collection | No software was used to collect data. Data were sequenced from PacBio Sequel and Illumina NovaSeq 6000. |
|---|---|
| Data analysis | We used publicly available and appropriately cited software as described. No commercial software or code was used in this study. Software are listed as follows: Canu (v1.5), Pilon (version 1.22), IrysSolve (v3.5_12162019, https://bionanogenomics.com/support/software-downloads/), PBJelly (version 15.8.24), BWA (version 0.7.10-r789), HiC-Pro (v2.8.1), LACHESIS (v1.0), BUSCO (version 5.2.0), Trinity (v2.8.5), BLAST (v 2.12.0+), RepeatScout (version 1.0.5), LTR-FINDER (version 1.05), MITE-hunter (version 1.0), PILER-DF (version 1.0), REPET (version 2.5), RepeatMasker (version 4.0.5), GeMoMa (version 1.3.1), GlimmerHMM (version 3.0.4), HISAT2 (version 2.0.4), Stringtie (version 1.2.3), BLAT (v.36), PASA (version 2.0.4), TopHat (version 2.0.12), Cufflinks (version 2.2.1), Transdecoder (version 2.0), EVidenceModeler (version 1.1.1), Blast2GO (version 4.1.8), GMAP (version 2015-06-12), OrthoFinder (v2.5.2), EDTA (v1.9.4), Cd-hit (v4.8.1), quota-alignment (version 1.0), MUSCLE (version 3.8.31), phyML (version v3.3.20190909), PAML (version 4.7b), MUMmer (version 4.0.0beta2), SVMU (v0.4-alpha), SyRI (v1.2), minimap2 (v2.21-r1071), SURVIVOR (v1.0.6), R (v4.03), vg (v1.38.0), GATK (v4.1.4.1), PLINK (v1.9.0b4.6), EMMAX (v20120210), Genetic type 1 Error Calculator (v0.2), EMBOSS package (v6.6.0).<br>Custom codes and scripts are available at https://github.com/HongboDoll/TomatoSuperPanGenome and https://doi.org/10.5281/zenodo.7396707. |

For manuscripts utilizing custom algorithms or software that are central to the research but not yet described in published literature, software must be made available to editors and reviewers. We strongly encourage code deposition in a community repository (e.g. GitHub). See the Nature Research guidelines for submitting code & software for further information.

## Data

Policy information about availability of data

All manuscripts must include a data availability statement. This statement should provide the following information, where applicable:

- Accession codes, unique identifiers, or web links for publicly available datasets
- A list of figures that have associated raw data
- A description of any restrictions on data availability

All assembled genome sequences and their annotation are publicly accessible through our database (http://caastomato.biocloud.net). We have also deposited the genome assemblies in the NCBI GenBank under the accession number PRJNA809001. Raw PacBio data, transcriptome and Hi-C sequencing reads have been deposited into NCBI sequence read archive (SRA) (https://www.ncbi.nlm.nih.gov/sra/) under BioProject accession number PRJNA756391. Whole-genome sequencing data were downloaded from NCBI (BioProjects: PRJNA259308, PRJNA353161, PRJNA454805 and PRJEB5235). The RepBase database was downloaded from https://www.girinst.org/server/RepBase/index.php.

# Field-specific reporting

Please select the one below that is the best fit for your research. If you are not sure, read the appropriate sections before making your selection.

☒ Life sciences          ☐ Behavioural & social sciences          ☐ Ecological, evolutionary & environmental sciences

For a reference copy of the document with all sections, see nature.com/documents/nr-reporting-summary-flat.pdf

# Life sciences study design

All studies must disclose on these points even when the disclosure is negative.

| | |
|---|---|
| Sample size | We selected 11 tomato accessions, representing nine wild and one cultivated tomato species. The logic of this selection was based on the extant wild (12) and cultivated (1) tomato species that are collectible. |
| Data exclusions | No samples were excluded in this study. Filters applied to eliminate low-quality sequencing data and genetic variants were properly described in the Methods section. |
| Replication | Three biological replicates with two technical replicates were used in the qRT-PCR experiment. Three independent T2 transgenic lines were generated for the estimation of single fruit weight, transverse diameter, longitudinal diameter, total fruit number and total fruit weight, in which three independent wild-type plants were also measured. All replications were successful and were used. |
| Randomization | For each tomato individual, the sampling process for genome DNA/RNA sequencing was randomly conducted. All WT and transgenic plants were exposed to the same growth condition and treatment. |
| Blinding | Blinding is not necessary for genome sequencing and assembly, since the investigators know which tomato species they were handling. The investigators were blinded to group allocation during collecting data from WT and transgenic tomato plants. |

# Reporting for specific materials, systems and methods

We require information from authors about some types of materials, experimental systems and methods used in many studies. Here, indicate whether each material, system or method listed is relevant to your study. If you are not sure if a list item applies to your research, read the appropriate section before selecting a response.

## Materials & experimental systems

| n/a | Involved in the study |
|---|---|
| ☒ | ☐ Antibodies |
| ☒ | ☐ Eukaryotic cell lines |
| ☒ | ☐ Palaeontology and archaeology |
| ☒ | ☐ Animals and other organisms |
| ☒ | ☐ Human research participants |
| ☒ | ☐ Clinical data |
| ☒ | ☐ Dual use research of concern |

## Methods

| n/a | Involved in the study |
|---|---|
| ☒ | ☐ ChIP-seq |
| ☒ | ☐ Flow cytometry |
| ☒ | ☐ MRI-based neuroimaging |

