## [Peer Review File · Nature Genetics]

Peer Review Information

Manuscript Title: Super-pangenome analyses highlight genomic diversity and structural variation across wild and cultivated tomato species

Corresponding author name(s): Professor Qinghui Yu, Hongbo Li, Professor Huan Wang

Reviewer Comments & Decisions:

Decision Letter, initial version:

6th Jan 2022

Dear Professor Yu,

Your Article, "The Super-pangenome Reveals the Panorama of Evolution and Structural Variation across Tomato Species" has now been seen by 3 referees. You will see from their comments copied below that while they find your work of considerable potential interest, they have raised quite substantial concerns that must be addressed. In light of these comments, we cannot accept the manuscript for publication, but would be very interested in considering a revised version that addresses these serious concerns.

We hope you will find the referees' comments useful as you decide how to proceed. If you wish to submit a substantially revised manuscript, please bear in mind that we will be reluctant to approach the referees again in the absence of major revisions.

To guide the scope of the revisions, the editors discuss the referee reports in detail within the team, with a view to identifying key priorities that should be addressed in revision. In this case, we think all referees have provided constructive reviews aimed at strengthening the analyses and the data interpretation, and we particularly ask that you perform additional analyses to fully address their technical comments and to strengthen your conclusions, compare your study with those published tomato pan-genomes and provide fair acknowledgement of other work in this area, and improve the language to avoid exaggeration and hyperbole. We hope that you will find the prioritized set of referee points to be useful when revising your study.

If you choose to revise your manuscript taking into account all reviewer and editor comments, please highlight all changes in the manuscript text file. At this stage we will need you to upload a copy of the manuscript in MS Word .docx or similar editable format.

We are committed to providing a fair and constructive peer-review process. Do not hesitate to contact us if there are specific requests from the reviewers that you believe are technically impossible or

unlikely to yield a meaningful outcome.

*2) If you have not done so already please begin to revise your manuscript so that it conforms to our Article format instructions, available [here](http://www.nature.com/ng/authors/article_types/index.html). Refer also to any guidelines provided in this letter.

[redacted]

If you wish to submit a suitably revised manuscript we would hope to receive it within 6 months. If you cannot send it within this time, please let us know. We will be happy to consider your revision so long as nothing similar has been accepted for publication at Nature Genetics or published elsewhere. Should your manuscript be substantially delayed without notifying us in advance and your article is eventually published, the received date would be that of the revised, not the original, version.

Thank you for the opportunity to review your work.

Sincerely,
Wei

Wei Li, PhD
Senior Editor
Nature Genetics
New York, NY 10004, USA
www.nature.com/ng

Reviewers' Comments:

Reviewer #1:

Remarks to the Author:

The manuscript describes the assembly and annotation of 11 diverse tomato lines and the construction of a pangenome. The pangenome was used to characterise diversity and combined with genotyping a larger population, identified structural variants associated with traits.

The methods follow the standard whole genome and assembly comparison pangenome approach and used the most relevant current methods for genome assembly. With some caveats, the analysis appears to be robust and the findings of interest, particularly the variation that causes almost doubling of yield. Given the importance of this trait I would have liked to have seen more discussion on how this could be applied to improve modern varieties.

The use of English could be improved and in particular, avoidance of generic statements, exaggeration and hyperbole. More scientific and unbiased language would enhance readability. Similarly, a fair acknowledgment of other work in this area would add balance.

The website presented seems unfinished, buggy and with some parts such as the browsers working extremely slowly if at all. It is good that the data is available for download, though the authors should confirm that the assemblies are also deposited in the relevant international repository.

Line 209, 16% of gene families are accession specific. This seems to be very high and would require some validation/support along with an explanation as to how this figure was derived.

Line 219 states that 3441 of the 4874 non-reference genes in the previously reported tomato pangenome are captured in this pangenome assembly. Where are the missing 1433 genes? Are they real genes or some artefact? This result combined with the 16% reported on line 209 makes me suspect the quality of the annotations or comparison of annotations.

Line 235 states that 18.4% of variations are in genes and suggests this is a low number. However the genic regions make up a small percentage of the genome and so 18.4% would suggest that there is a higher density of variants in genes than in the intergenic space. This contradicts all that is known

about genome variation.

Line 249, what would the difference between hyperactive and active TEs be?

Line 312, why are SVs proved to be a type of 'crucial' variant? Saying that SVs have a 'more significant impact' on phenotypes is a gross generalisation.

Line 319, how were the SVs genotyped across the 321 accessions?

Line 339, finding a significant association with SV that could not be detected using SNPs may be due the difference in the number of these markers. A comparison should be made using an equivalent subset of evenly distributed SNP markers to truly demonstrate that SVs have greater power in GWAS compared to SNPs. I would expect that the distribution and number of markers has a greater influence on association than the type of marker.

Line 346, 'upper and lower ends' should be defined more precisely, how do these relate to gene density?

Line 363, 'great values' should be quantified

Line 368 suggests that other types of pangenome assembly cannot be used for read mapping and variant calling. This is clearly untrue from the numerous studies that have performed this.

Reviewer #2:

Remarks to the Author:

This manuscript presented an impressive dataset with 11 tomato high quality assemblies, including wild and cultivated strains. The rich data will definitely promote tomato biological and evolutionary studies, also will improve tomato breeding. Unfortunately, the analyses in this version do not really do the dataset justice, to help to enhance the quality of this manuscript, my main criticism as follow: Overall, given the paper (Alonge et al., Cell, 2020) had published a tomato pan-genome with ~100 accessions, and most of those are cultivated accessions, therefore, this study must emphasize and show the justice of wild species in the tomato evolutionary and/or biological studies from different aspects.

1: I did not find the result about the heterozygosity rate of wild species, it was supposed to be basic analysis for wild species. If heterozygosity rate was high, then the genome assembly in this study might need to be reconsidered, or at least mention how deal with heterozygous alleles in following analysis.

2: Given that this study provided 9 wild species of tomato, this was a quite good opportunity to use the rich data to show us about artificial and natural selection during tomato evolution and domestication process, by coupling with other public datasets, such as the dataset in the study (Alonge et al., Cell, 2020).

3: The content of "Transposons drive genome evolution in Solanum" section was unsurprised for me, so it was not necessary to separate these result to an independent section.

4: Regarding the section "Structural variants among cultivated and wild tomatoes", the authors mentioned that a number of SNP/Indels in this dataset potentially influenced several genes, however, the authors didn't further confirm these variations, or didn't analyze what these results could indicate

regarding artificial or natural selection. Moreover, I did not find any description of calling SNP/Indel in the methods.

About SV, the author found that insertions and deletions were more likely to be found at both ends of the chromosomes, what the result could suggest? Was this lead by some inappropriate technical false? because I did not find similar pattern for SVs distribution along chromosome in (Alonge et al. Cell, 2020).

Regarding SV analysis at genome-wide scale, the author just described "We identified 5,186 large insertions or deletions (> 50 bp) present either in all wild or all domesticated tomato genomes investigated in this study, some of which led to insertion of protein-coding genes present only in wild tomatoes". It did not do the dataset justice again. For example, it could be easy to merge the SVs in the study (Alonge et al. Cell, 2020), to comprehensively analysis the SV distribution in different populations. Also, it is easy to infer the ancestry state of SVs, such analysis will definitely enhance the quality of this study.

Additionally, I have no idea why the author mentioned the 7.1Mb inversion, was this inversion important for any traits? Also, I have no idea why the author mentioned "For the fruit flavor-related TomLoxC gene9, we found that *S. pennellii*, *S. habrochaites*, *S. chilense* and *S. neorickii* had the genotype that contributes to a desirable tomato flavor.", what genotype? what did this sentence want to illuminate?

The analysis about the content "to investigate whether SVs might affect nearby gene expression...." was very rough, moreover, it was not appropriate that the author only used two accessions for the correlation analysis between SV and expression level.

5: Regarding the section "Graph-based genome enables SV-based genome-wide association studies (GWAS) in tomato", firstly, the basic description about graph-based genome was absence. About the GWAS using SV and SNP, although the author tried to use several examples to illuminate how powerful the SV-based GWAS was in capturing variations responsible for different traits, the analysis for each example was very rough. For example, the authors showed that Sgal04g002480 was possibly involved in regulating metabolism of malic acids, the key information (the variation of Sgal04g002480, and its influence on protein coding or gene expression) were absent.

I have no idea why the author expanded their association panel to 362 annotated metabolites, moreover, they just show "272 signals significantly associated with 89 metabolites", then no more any other analysis about these results. Therefore, I think that the expanded results were redundant, no any help for this manuscript.

6: For "discussion" section, it was worthy to discuss some disadvantages in this study.

Reviewer #3:

Remarks to the Author:

In this study, Li et al. sequenced and de novo assembled chromosome-scale genomes of nine wild species and two cultivated accessions of tomato, and constructed a panSV-genome for tomato. They claimed that this study potentially provides valuable resources for tomato functional studies and breeding. However, in-depth analysis and experimental validation are definitely required to support the conclusions. In the context that several high-profile publications have recently reported the pangenomes of tomato (Gao et al., Nat Genet. 2019, 51(6):1044-1051.; Alonge et al., Cell. 2020, 182(1):145-161.), the innovation of this research over those published articles should be analyzed and provided. In addition, several major points need to be addressed.

1. Compared to the other ten tomato genomes, the wild tomato *S. galapagense* 'LA0436' showed a higher level of completeness in term of genome assembly. Is this result related to the use of its

- specific Bionano data? The authors mentioned that 'LA0436' has high salt tolerance, however, the other accessions, such as 'LA2951' and 'LA1969', also exhibit high stress and disease tolerance. Why the Bionano data was not applied for the other ten tomato genomes? More importantly, does the different assembly level of tomato genomes influence the following evolution and structural analyses? For example, it is widely accepted that *S. pimpinellifolium* is the ancestor of cultivated tomatoes, however the current study showed that *S. galapagense* is closer than *S. pimpinellifolium* to cultivated tomatoes. Does this discrepancy relevant to the different quality of the genome assembly?
2. The number of core gene sets are greatly different from the pan genome previously reported. The authors should conduct an in-depth comparison between their core gene sets with those of the published super-pangenomes. What is the similarity of the genes in the core gene set? Authors should highlight the innovations of their work over the recently published tomato pan-genomes.
 3. The authors divided the 12 Lycopersicon genomes into four clades (Fig 1a) and also identified genome-specific genes by de novo annotation. I wonder whether they could identify clade-specific genes? If yes, what's the function of these clade-specific genes?
 4. Line 260, the authors listed some SVs that have been verified by researchers using their super-pangenome data. I wonder if they also found the CNV in fw3.2 gene controlling fruit size in the super-pangenome data.
 5. Line 316-319 mentioned "Here, we constructed a tomato graph-based genome by integrating the linear reference genome sequence of *S. galapagense* and a total of 88,817 SVs (41,063 insertions, 46,390 deletions and 1,364 inversions) identified from 12 tomato genomes.", but the numbers of SVs here are different from the SV identified at Line 245-247. Are the two sets of SVs identified by different methods? Please clarify the criteria used to identify SVs.
 6. Authors mentioned that a pan-SV map has been built from Oxford Nanopore long reads of 100 diverse tomato lines by Lippman Lab (Alonge et al., Cell. 2020, 182(1):145-161.). They should compare the reported panSV-genome with the graph-based genome built in this work. They should also evaluate whether the SVs they identified is close to saturation.
 7. The 321 tomato lines used in GWAS only include modern, heirloom, and wild accessions of the tomato (*Solanum lycopersicum*) and its closest relative, *S. pimpinellifolium*. Please combine the previously reported panSV-genome with the SV identified here to generate a merged graph-based genome. Given that most of this set of phenotypic data are derived from modern varieties, with only one or two closely-related wild species, I doubt that these data sets are suitable to be used to assess the value of your pan-genome.
 8. All codes and scripts used in this study should be deposited into a public platform or website, such as Github

Author Rebuttal to Initial comments

Dear Dr. Wei Li,

We greatly appreciate your dedication and that of the reviewers to help us improve the manuscript. According to the comments, we performed additional analyses and substantially revised the manuscript, highlighting the innovation of this study compared with existing research, as well as presenting fair acknowledgement of other work on tomato genomics. Below we provide a point-by-point response to the reviewers' comments and indicate how we have modified the manuscript. All revision regarding reviewer's concerns has been highlighted in **yellow** background.

Sincerely,

Qinghui Yu, on behalf of all co-authors who agreed on this submission

Reviewers' Comments:

Reviewer #1:

Remarks to the Author:

The manuscript describes the assembly and annotation of 11 diverse tomato lines and the construction of a pangenome. The pangenome was used to characterise diversity and combined with genotyping a larger population, identified structural variants associated with traits.

Response: Thanks for the comments.

The methods follow the standard whole genome and assembly comparison pangenome approach and used the most relevant current methods for genome assembly. With some caveats, the analysis appears to be robust and the findings of interest, particularly the variation that causes almost doubling of yield. Given the importance of this trait I would have liked to have seen more discussion on how this could be applied to improve modern varieties.

Response: Thanks for the comments. We have added discussion on the potential of the newly identified wild-type allele to increase yield in modern tomato varieties. Please see the revised manuscript (Lines 425-446).

The use of English could be improved and in particular, avoidance of generic statements, exaggeration and hyperbole. More scientific and unbiased language would enhance readability. Similarly, a fair acknowledgment of other work in this area would add balance.

Response: Thanks for the comments. We have substantially revised the manuscript and rephrased the statements with exaggeration and generic expressions, as well as added comparison with other studies in the field of tomato genomics, such as Alonge *et al.*¹ and Gao *et al.*².

The website presented seems unfinished, buggy and with some parts such as the browsers working extremely slowly if at all. It is good that the data is available for download, though the authors should confirm that the assemblies are also deposited in the relevant international repository.

Response: Thanks for the comments. We have fixed bugs in the website and addressed the network connection issue. We have also deposited the genome assemblies in the NCBI GenBank under the accession number PRJNA809001, which will be released upon publication. This information has been added into the **Data availability** section.

Line 209, 16% of gene families are accession specific. This seems to be very high and would require some validation/support along with an explanation as to how this figure was derived.

Response: Thanks for the comments. Since the determination of accession-specific gene families would be skewed owing to unannotated genes, a common issue during whole-genome gene prediction, we have handled this issue and re-performed pan-genome analysis. We first aligned coding sequences of all predicted genes to each of the 13 tomato genomes using GMAP³. If a gene showed both more than 80% of alignment coverage and identity with no gene predicted within the aligned regions, it was considered to be an unannotated gene in the corresponding genome and would not be regarded as “missing” in the further analysis. We then performed pan-genome analyses based on a Markov clustering approach using OrthoFinder⁴, resulting in 40,457 gene families. Based on their presence frequencies, we classified these gene families into three categories: core (those present in all 13 individuals, 21,838, 54.0%), dispensable (those present in 2-12 individuals, 15,539, 38.4%) and accession-specific (those present in only one accession, 3,080, 7.6%). Considering that our super pan-genome contains ten species within the *Solanum* genus, this proportion of 7.6% is comparable with a previous study of ten *Oryza* species, which observed that an average of 6% of annotated loci per species were species-specific⁵. These results and relevant methods have been updated in the revised manuscript for the section “**Super-pangenome of tomato**”.

Line 219 states that 3441 of the 4874 non-reference genes in the previously reported tomato pangenome are captured in this pangenome assembly. Where are the missing 1433 genes? Are they real genes or some artefact? This result combined with the 16% reported on line 209 makes me suspect the quality of the annotations or comparison of annotations.

Response: We did not anticipate that all the 4,874 non-reference genes derived from 586 tomato accessions² could be captured by our super pan-genome, since only 13 representative tomato accessions from 11 *Solanum* species were sampled in this study, among which only five were from the species overlapping with those of the 586 tomato accessions². We used a universal, widely accepted pipeline for gene prediction for all the 13 accessions by integrating transcriptome evidence, *ab initio* gene model prediction and homologous protein alignment approaches. The high completeness of our gene prediction

was supported by a mean BUSCO score of 93.6%. Regarding the comparison between gene prediction dataset, we aligned the amino acid sequences of the 4,874 non-reference genes to the genome sequences of the 13 tomato accessions to check their presence. This approach avoids potential falsely predicted genes in these 13 genomes. Therefore, the 1,433 missing should be genes absent in our super pan-genome.

Line 235 states that 18.4% of variations are in genes and suggests this is a low number. However the genic regions make up a small percentage of the genome and so 18.4% would suggest that there is a higher density of variants in genes than in the intergenic space. This contradicts all that is known about genome variation.

Response: Thanks for the comments. We have calculated the variant density within genic and inter-genic regions and found that they were comparable (0.063 and 0.066 variants per bp, respectively). To be clearer, we have rephrased this sentence to “Most of these variants (81.6%) were localized in intergenic regions, whereas 18.4% resided within genes, showing a similar variant density (0.066 and 0.063 variants per bp in intergenic and gene regions, respectively).” (Lines 236-238).

Line 249, what would the difference between hyperactive and active TEs be?

Response: We have rephrased “hyperactive” to “active” in corresponding descriptions, as used in relevant literature.

Line 312, why are SVs proved to be a type of ‘crucial’ variant? Saying that SVs have a ‘more significant impact’ on phenotypes is a gross generalisation.

Response: We have rephrased this sentence to “Numerous studies have suggested that SVs are causative variants responsible for agronomically important traits^{1,2,6,7}”.

Line 319, how were the SVs genotyped across the 321 accessions?

Response: By mapping the Illumina resequencing data of the 321 accessions to the constructed graph-based genome, the state (including read pair and split read information) and coverage of mapped reads around the SV breakpoints were examined and genotypes of each SVs were assigned for the 321 individuals. The genotyped SVs with fewer than two supporting reads were removed. We have added this into the revised manuscript in the **Methods** section (Lines 672-677).

Line 339, finding a significant association with SV that could not be detected using SNPs may be due the difference in the number of these markers. A comparison should be made using an equivalent subset of evenly distributed SNP markers to truly demonstrate that SVs have greater power in GWAS compared to SNPs. I would expect that the distribution and number of markers has a greater influence on association than the type of marker.

Response: Since the number of SNP markers was remarkably higher than that of SV markers (3,148,478 versus 33,933 used for GWAS), one would expect that association signals are more likely to be detected by SNPs. We observed that the distance between 99.98% of SVs (33,925) and their closest SNPs (upstream or downstream) was fewer than 15 kb (**Response Fig. 1**), much smaller than the distance of linkage disequilibrium decay in cultivated tomatoes⁸ (865.7 kb in big-fruited tomatoes and 256.8 kb in cherry tomatoes when r^2 declined to 0.2). Furthermore, among the 495 SV associations which could not be detected by SNPs, 490 showed distances < 15 kb when compared with their nearest SNPs (**Response Fig. 1**). We, therefore, argue that the identification of SV association signals which could not be detected by SNPs was not likely to be impacted by the density of different markers. Nonetheless, we emphasized in the revised manuscript that the SV-based GWAS serves as an important complement to the SNP-based GWAS by identifying numerous additional association peaks (**Lines 398-401 and Lines 413-415**).

Response Fig. 1. Distance distribution between SVs and their closest SNPs. Green lines indicate SNPs downstream of SVs while yellow lines denote SNPs upstream of SVs. X-axis represents the distance between SV and its nearest flanking SNP.

Line 346, ‘upper and lower ends’ should be defined more precisely, how do these relate to gene density?

Response: We have substantially rewritten this section and removed this sentence. Please see the revised manuscript (**Lines 385-415**).

Line 363, 'great values' should be quantified

Response: We have rephrased this sentence to “we identified a wild-tomato allele that can increase fruit yield by an average of 67.1% in overexpression transgenic tomatoes (**Fig. 3d-f**)” (**Lines 428-429**).

Line 368 suggests that other types of pangenome assembly cannot be used for read mapping and variant calling. This is clearly untrue from the numerous studies that have performed this.

Response: We agree with the reviewer and have removed the sentence “moreover, these pan-genomes can hardly be utilized similar to a traditional linear reference, such as performing read mapping and variant calling”.

Reviewer #2:

Remarks to the Author:

This manuscript presented an impressive dataset with 11 tomato high quality assemblies, including wild and cultivated strains. The rich data will definitely promote tomato biological and evolutionary studies, also will improve tomato breeding. Unfortunately, the analyses in this version do not really do the dataset justice, to help to enhance the quality of this manuscript, my main criticism as follow:

Overall, given the paper (Alonge et al., Cell, 2020) had published a tomato pan-genome with ~100 accessions, and most of those are cultivated accessions, therefore, this study must emphasize and show the justice of wild species in the tomato evolutionary and/or biological studies from different aspects.

1: I did not find the result about the heterozygosity rate of wild species, it was supposed to be basic analysis for wild species. If heterozygosity rate was high, then the genome assembly in this study might need to be reconsidered, or at least mention how deal with heterozygous alleles in following analysis.

Response: We are sorry about that. The heterozygosity issue has been handled during the genome assembly step. We have added results of genome heterozygosity in the **Results** section (**Lines 109-111 and Lines 125-129**) as well as more details in the **Methods** section (**Lines 531-535**) of the revised manuscript. We first calculated the heterozygosity rate of all nine wild tomato species based on k -mer ($k = 19$) analyses using Illumina reads (**Response Fig. 2**) and found that *S. corneliomulleri* (0.98%), *S. peruvianum* (0.66%) and *S. chilense* (0.65%) displayed relatively higher levels of heterozygosity (> 0.5%).

Response Fig. 2. K-mer frequency distribution of genomes of the nine selected wild tomato species. Estimated heterozygosity rates for *S. corneliomulleri*, *S. peruvianum* and *S. chilense* are also marked.

The purge of putative haplotigs in the initially assembled genomes of the nine wild tomato species was leveraged by the Hi-C data. Briefly, we first mapped Hi-C reads to the assembled contigs using BWA⁹ (command “aln”), and only retained uniquely mapped read pairs with mapping quality > 20. These results were then fed into HiC-Pro¹⁰ to identify valid interaction pairs, generating the contact matrix. Based on the matrix, we next applied LACHESIS¹¹ to cluster and order the assembled contigs into pseudo-chromosomes and manually checked the Hi-C interaction heat maps to identify potential genomic regions containing haplotigs due to heterozygosity, which were excluded from the assembly. The manual curation step was re-performed several times, until the chromatin interaction signals reflecting putative haplotigs were undetectable.

We also mapped Hi-C reads to their genomes and found no chromatin interaction pattern representing two un-collapsed haplotypes (**Response Figs. 3-5**). These results indicate monoploid assembled genomes.

Response Fig. 3. Hi-C interaction heat map of *S. corneliomulleri* with 200-kb resolution.

Response Fig. 4. Hi-C interaction heat map of *S. peruvianum* with 200-kb resolution.

Response Fig. 5. Hi-C interaction heat map of *S. chilense* with 200-kb resolution.

2: Given that this study provided 9 wild species of tomato, this was a quite good opportunity to use the rich data to show us about artificial and natural selection during tomato evolution and domestication process, by coupling with other public datasets, such as the dataset in the study (Alonge et al., Cell, 2020).

Response: Thanks for the comment. We are sorry that the currently available data is not sufficient to perform analyses of natural selection from distantly related wild tomato species (*S. chmielewskii*, *S. neorickii*, *S. corneliomulleri*, *S. peruvianum*, *S. chilense*, *S. habrochaites* and *S. lycopersicoides*) to the cultivated tomato, as this would require genome sequencing data from multiple individuals of each of the wild species. Unfortunately, this type of dataset is only available for the wild progenitor of cultivated tomatoes, *S. pimpinellifolium*. Regarding artificial selection, we think it is not necessary to conduct relevant analyses, since the genomic basis underlying tomato domestication from *S. pimpinellifolium* to *S. lycopersicum* has already been comprehensively characterized by several studies using both SNPs and

SVs from population-scale genome resequencing^{2,8,12}, all of which have collected more complete datasets of *S. pimpinellifolium* and *S. lycopersicum* accessions than did this study.

To emphasize the innovation of the nine wild tomato genomes assembled in this study, we have integrated the SV dataset from Along *et al.* to investigate how SVs have diverged during the evolution from distantly related wild species to cultivated tomato accessions. Please see the revised manuscript (Lines 279-344).

3: The content of “Transposons drive genome evolution in Solanum” section was unsurprised for me, so it was not necessary to separate these result to an independent section.

Response: We agree with the reviewer and have integrated this section into the section “**Eleven wild and cultivated tomato reference genomes**”. Please see the revised manuscript (Lines 145-159).

4: Regarding the section “Structural variants among cultivated and wild tomatoes”, the authors mentioned that a number of SNP/Indels in this dataset potentially influenced several genes, however, the authors didn’t further confirm these variations, or didn’t analyze what these results could indicate regarding artificial or natural selection. Moreover, I did not find any description of calling SNP/Indel in the methods.

Response: Thanks for the comments. We have conducted PCR product sequencing to verify those variants marked in **Supplementary Figs. 10-15**, and all of them were verified. We have also added description of SNP/InDel calling approaches in the **Methods** section (Lines 617-621).

About SV, the author found that insertions and deletions were more likely to be found at both ends of the chromosomes, what the result could suggest? Was this lead by some inappropriate technical false? because I did not find similar pattern for SVs distribution along chromosome in (Alonge et al. Cell, 2020).

Response: The SVs identified in this study were more likely to distribute at both ends of the chromosomes, corresponding to euchromatin regions with higher density of genes. This pattern is consistent with a recent study that identified SVs between *S. pimpinellifolium* LA2093 and *S. lycopersicum* Heinz 1706 (ref.¹³). We also checked **Fig. 2A** in Alonge *et al.*¹ and did find that the similar SV distribution pattern was also present in their dataset for the wild tomato accession LA1589 (*S. pimpinellifolium*): SVs were more likely to distribute at both ends of 11 out of the 12 chromosomes (chr1, chr3, chr4, chr5, chr6, chr7, chr8, chr9, chr10, chr11 and chr12) (**Response Fig. 6**). We did not observe a

high SV density in the upper arm of chromosome 2, possibly due to the presence of dense 45S rDNA repeat¹⁴.

Response Fig. 6. Comparison of the density of SVs identified in this study and Along *et al.*¹ a, Distribution of structural variants from the 12 genomes across the 12 tomato chromosomes. b, Circos plot (bottom) depicts genome-wide SV frequency for five notable accessions. Fig. b was adopted from Along *et al.*¹.

Regarding SV analysis at genome-wide scale, the author just described “We identified 5,186 large insertions or deletions (> 50 bp) present either in all wild or all domesticated tomato genomes investigated in this study, some of which led to insertion of protein-coding genes present only in wild tomatoes”. It did not do the dataset justice again. For example, it could be easy to merge the SVs in the study (Alonge *et al.* Cell, 2020), to comprehensively analysis the SV distribution in different populations. Also, it is easy to infer the ancestry state of SVs, such analysis will definitely enhance the quality of this study.

Response: Thanks for the comments. We have downloaded the SV set from Alonge *et al.*¹ and compared their SVs with those identified in this study. We then merged them with our SV set and performed SV distribution analyses among different tomato populations, as well as analyzed the alteration of SV presence frequencies during tomato evolution. These results have been arranged into a new section “**Hidden genetic diversity unveiled by genomes of tomato wild species**”, in which we provide additional insights into SV evolution among distantly related tomato wild species, as well as offer a useful

dataset for further characterizing gene underlying phenotypes greatly diverged between wild and cultivated tomatoes. Please see the revised manuscript for details (**Lines 279-344**).

Additionally, I have no idea why the author mentioned the 7.1Mb inversion, was this inversion important for any traits? Also, I have no idea why the author mentioned “For the fruit flavor-related *TomLoxC* gene⁹, we found that *S. pennellii*, *S. habrochaites*, *S. chilense* and *S. neorickii* had the genotype that contributes to a desirable tomato flavor.”, what genotype? what did this sentence want to illuminate?

Response: We are sorry for the confusion. This inversion is not associated with agronomic traits. Large inversions have been reported to suppress recombination by reducing crossing-over^{15,16}, resulting in severe linkage drag when conducting backcross breeding. To avoid this, it is necessary to select donor lines without inverted fragments harboring target genes. Based on the 11 high-quality chromosome-scale tomato genomes, we identified 12 (*S. lycopersicum* cv. Heinz 1706) - 42 (*S. chmielewskii*) megabase-scale inversions (**Supplementary Table 19**). The 7.1-Mb inversion was thus an example displaying a clear divergence signature between wild and domesticated tomatoes: it was present in all species of clade IV (*S. lycopersicum*, *S. galapagense* and *S. pimpinellifolium*) but absent in seven out of eight wild species. Hence, when breeders attempt to introduce genes from wild tomatoes within this 7.1-Mb segments into elite cultivars by backcrossing, *S. pennellii* would be an ideal donor parent, since it does not carry inversion within this region. Equipped with the map of large-scale inversions and genome information of these wild tomato species, breeders are now able to preclude putative linkage drag when applying backcross breeding, based on appropriate selection of donor and acceptor lines. These results have been integrated into the revised manuscript (**Lines 279-296**), to highlight the innovation of the genome sequences for nine wild tomato species presented in this study.

Regarding the *TomLoxC* locus, we initially wanted to know which wild species contained the favorable upstream alleles of *TomLoxC*, a gene contributing to desirable fruit flavor. This information could provide guidance for selecting donor parent in backcrossing breeding. Therefore, we have rephrased this sentence to “Two different alleles (4,724-bp and 4,151-bp) have been identified at 149 bp upstream of *TomLoxC*, a gene encoding a 13-lipoxygenase, and the 4,151-bp allele was reported to contribute to desirable fruit flavor and is rare in cultivated tomatoes². We found that *S. pennellii*, *S. habrochaites*, *S. chilense* and *S. neorickii* carried the 4,151-bp allele upstream of *TomLoxC*, suggesting that these wild species have the potential to improve fruit flavor in cultivated tomato by backcrossing”. (**Lines 269-275**)

The analysis about the content “to investigate whether SVs might affect nearby gene expression...” was very rough, moreover, it was not appropriate that the author only used two accessions for the correlation analysis between SV and expression level.

Response: We agree with the Reviewer and have removed this result in the revised manuscript.

5: Regarding the section “Graph-based genome enables SV-based genome-wide association studies (GWAS) in tomato”, firstly, the basic description about graph-based genome was absent. About the GWAS using SV and SNP, although the author tried to use several examples to illuminate how powerful the SV-based GWAS was in capturing variations responsible for different traits, the analysis for each example was very rough. For example, the authors showed that Sgal04g002480 was possibly involved in regulating metabolism of malic acids, the key information (the variation of Sgal04g002480, and its influence on protein coding or gene expression) were absent.

Response: We have added a brief description of graph-based genome: “Graph-based genomes are capable of storing both reference and alternative allele sequences, facilitating mapping of short reads from SV regions and thus SV genotyping^{17,18}” in the revised manuscript (**Lines 392-394**). Regarding the detailed analysis of candidate genes from GWAS, in the revised manuscript, we focused on association signals where SVs showed more significant correlation with given phenotypes than did SNPs, instead of candidate gene isolation, since it is quite difficult to predict candidate genes merely by GWAS without fine mapping or functional experimental validation in tomato, a species with long linkage disequilibrium⁸ (865.7 kb in big-fruited tomatoes and 256.8 kb in cherry tomatoes when r^2 declined to 0.2). Nevertheless, the identified GWAS peaks, displaying significant correlation with fruit flavor-related metabolites, will also be useful in downstream analyses, such as candidate gene mapping and SV-based breeding selection. Meanwhile, we also emphasized in the revised manuscript that the SV-based GWAS serves as an important complement to the SNP-based GWAS by identifying numerous additional association peaks. Therefore, the section “**Graph-based genome enables SV-based genome-wide association studies (GWAS) in tomato**” has been rephrased. Please see the revised manuscript for details (**Lines 385-415**). We have also added discussion regarding difficulty of identifying casual genes from GWAS in tomato (**Lines 469-471**).

I have no idea why the author expanded their association panel to 362 annotated metabolites, moreover, they just show “272 signals significantly associated with 89 metabolites”, then no more any other analysis about these results. Therefore, I think that the expanded results were redundant, no any help for this manuscript.

Response: To be clearer, we have integrated the results from the 362 fruit metabolites and the 32 fruit flavor-related compounds. We presented significantly associated SVs with these metabolites, which showed lower p -values than SNPs, serving as potential markers in breeding selection (**Lines 385-415**).

6: For “discussion” section, it was worthy to discuss some disadvantages in this study.

Response: We have added additional discussion regarding potential limitations of this study. Please see the revised manuscript (**Lines 457-471**).

Reviewer #3:

Remarks to the Author:

In this study, Li et al. sequenced and de novo assembled chromosome-scale genomes of nine wild species and two cultivated accessions of tomato, and constructed a panSV-genome for tomato. They claimed that this study potentially provides valuable resources for tomato functional studies and breeding. However, in-depth analysis and experimental validation are definitely required to support the conclusions. In the context that several high-profile publications have recently reported the pangenomes of tomato (Gao et al., Nat Genet. 2019, 51(6):1044-1051.; Alonge et al., Cell. 2020, 182(1):145-161.), the innovation of this research over those published articles should be analyzed and provided. In addition, several major points need to be addressed.

1. Compared to the other ten tomato genomes, the wild tomato *S. galapagense* 'LA0436' showed a higher level of completeness in term of genome assembly. Is this result related to the use of its specific Bionano data? The authors mentioned that 'LA0436' has high salt tolerance, however, the other accessions, such as 'LA2951' and 'LA1969', also exhibit high stress and disease tolerance. Why the Bionano data was not applied for the other ten tomato genomes? More importantly, does the different assembly level of tomato genomes influence the following evolution and structural analyses? For example, it is widely accepted that *S. pimpinellifolium* is the ancestor of cultivated tomatoes, however the current study showed that *S. galapagense* is closer than *S. pimpinellifolium* to cultivated tomatoes. Does this discrepancy relevant to the different quality of the genome assembly?

Response: Bionano data were only used for scaffolding (connecting the assembled contigs) and did not add any extra sequences to the assemblies, which, therefore, did not contribute to the higher level of completeness of the *S. galapagense* 'LA0436' assembly. Indeed, the higher completeness should be due to the higher amount of PacBio data used for assembling the LA0436 genome. The ten other tomatoes were also assembled into chromosome levels by leveraging Hi-C data, which are of sufficient quality for further analyses in this study; therefore, also considering the cost issue, we did not apply Bionano sequencing for additional scaffolding steps for the ten other tomatoes.

We agree with the Reviewer that other accessions sampled in our study also display high tolerance to stress and diseases, and have rephrased the sentence “We assembled a high-quality chromosome-scale reference genome of *S. galapagense* 'LA0436', a wild tomato native to Galapagos islands with high salt tolerance” to “We assembled a high-quality chromosome-scale reference genome of a wild tomato *S. galapagense* 'LA0436'”.

The different assembly qualities of these genomes will not impact the following evolution and structural analyses. 1) BUSCO evaluation indicated that the 11 assemblies had comparable BUSCO scores, reflecting similar completeness of genic regions (**Table 1**). Since the following phylogenetic

analyses were solely based on the single-copy orthologous genes, these will not be biased by the different assembly qualities. 2) Genome comparisons were conducted among the chromosome-level assemblies for all these genomes, and only structural variants (SVs) within a single contig (an assembled fragment without any gap sequence) were retained. Hence, our analyses of SVs will not be impacted by the genomes of different assembly continuity either.

We agree that *S. pimpinellifolium* is proposed to be the progenitor of cultivated tomatoes. Notwithstanding, The closer relationship of *S. galapagense* than *S. pimpinellifolium* to cultivated tomatoes has also been reported in several studies. 1) A more recent divergence time for *S. galapagense* and *S. lycopersicum* (0.19 MYA) than *S. pimpinellifolium* and *S. lycopersicum* (0.44 MYA) in Strickler *et al.* (**Table 2**)¹⁹. 2) *S. galapagense* accessions are evolutionarily closer to *S. lycopersicum* than *S. pimpinellifolium* accessions, which were supported in Strickler *et al.* (**Fig. 3**)¹⁹, Lin *et al.* (**Fig. 1b**)⁸, Gao *et al.* (**Fig. 2c**)² and Aflitos *et al.* (**Fig. 3**)¹². However, these results do not suggest that *S. galapagense* is the progenitor of cultivated tomatoes, since *S. galapagense* have evolved separately and been endemic in the Galapagos Islands^{20,21}.

2. The number of core gene sets are greatly different from the pan genome previously reported. The authors should conduct an in-depth comparison between their core gene sets with those of the published super-pangenomes. What is the similarity of the genes in the core gene set? Authors should highlight the innovations of their work over the recently published tomato pan-genomes.

Response: The number of core genes in our study (23,839) is lower than that of the previously reported pan-genome of 586 tomato accessions (29,938)². This difference could be due to the significantly higher divergence level among the 13 tomato accessions, including nine wild species, used in this study, since the tomato pan-genome constructed by Gao *et al.*² used 519 cultivated tomatoes and 67 closely related wild tomato accessions belonging to two species: *S. pimpinellifolium* and *S. galapagense*. This is supported by the drastically shorter branch length of cultivated tomatoes (including 12 accessions from Along *et al.*¹), as compared with that of the wild tomato species investigated here (**Response Fig. 7**).

Response Fig. 7. Phylogenetic relationships among the 22 cultivated and wild tomatoes. SLL, *S. lycopersicum*; SP, *S. pimpinellifolium*; SLC, *S. lycopersicum* var. *cerasiforme*

We have conducted in-depth comparison between the core genes identified in this study and those reported in Gao *et al.*² (Lines 194-203 and Lines 210-213). To highlight the innovation of this study, we have also compared the pan-genome constructed herein with that reported in Gao *et al.*², and identified 9,394 genes that are only present in our dataset, with follow-up analyses. Please see the section **Super-pangenome of tomato** in the revised manuscript for details (Lines 219-223).

3. The authors divided the 12 *Lycopersicon* genomes into four clades (Fig 1a) and also identified genome-specific genes by de novo annotation. I wonder whether they could identify clade-specific genes? If yes, what's the function of these clade-specific genes?

Response: As suggested by the Reviewer, we identified 1,363, 238, 104 and 1,274 genes which are uniquely present in clade I, II, III and IV, respectively. We found that genes specific to clade I displayed functions of e.g., regulation of photoperiodism and malate transport, while some clade IV (ref-fruited tomato clade)-specific genes were involved in biological processes such as protein transport and localization. These results have been integrated into the “**Super-pangenome of tomato**” section in the revised manuscript (Lines 205-209).

4. Line 260, the authors listed some SVs that have been verified by researchers using their super-pangenome data. I wonder if they also found the CNV in *fw3.2* gene controlling fruit size in the super-pangenome data.

Response: We have checked the identified SVs localized at the flanking regions of the *fw3.2* gene and, unfortunately, did not identify this CNV in our dataset.

5. Line 316-319 mentioned "Here, we constructed a tomato graph-based genome by integrating the linear reference genome sequence of *S. galapagense* and a total of 88,817 SVs (41,063 insertions, 46,390 deletions and 1,364 inversions) identified from 12 tomato genomes.", but the numbers of SVs here are different from the SV identified at Line 245-247. Are the two sets of SVs identified by different methods? Please clarify the criteria used to identify SVs.

Response: We are sorry for the confusion. We identified SVs by applying both SVMU²² and SyRI²³, and SVs produced by either two software were kept after a set of filters (please see the **Methods** for more details). In the original manuscript, we only retained SVs that were supported by both two of our SV calling approaches to construct a graph-based genome. Therefore, the number of SVs which were integrated into the graph-based genome was lower than that of identified SVs described in the previous section.

We, however, noticed that the previously defined criterion that only SVs supported by SVMU and SyRI were used to build the graph-based genome, was possibly too stringent. Therefore, also based on the comments from Reviewer #2, we have integrated the reported SVs from Alonge *et al.*¹ and all the identified SVs in our study to construct a graph-based genome, in the revised manuscript.

6. Authors mentioned that a pan-SV map has been built from Oxford Nanopore long reads of 100 diverse tomato lines by Lippman Lab (Alonge *et al.*, Cell. 2020, 182(1):145-161.). They should compare the reported panSV-genome with the graph-based genome built in this work. They should also evaluate whether the SVs they identified is close to saturation.

Response: We thank the Reviewer for pointing this out. As also suggested by the Reviewer #2, We have downloaded the SV set from Alonge *et al.*¹ and compared their SVs with those identified in this study. We then merged them with our SV set and performed SV distribution analyses among different tomato populations, as well as analyzed the alteration of SV presence frequencies during tomato evolution. These results have been arranged into a new section "**Hidden genetic diversity unveiled by genomes of tomato wild species**", in which we provide additional insights into SV evolution among distantly related tomato wild species, as well as offer a useful dataset for further characterizing gene underlying

phenotypes greatly diverged between wild and cultivated tomatoes. Please see the revised manuscript for details (**Lines 279-344**).

We have also performed a cumulative analysis of SV numbers when adding increasing tomato genomes. This suggested that SVs, in terms of insertions and deletions identified herein, did not reach a saturation (**Response Fig. 8**), due largely to the diverse nature of the wild species used in this study, with only one accession from each wild species being sampled.

Response Fig. 8. Cumulative number of SVs in terms of deletions (left) and insertions (right). Grey points represent the number of SVs when adding different numbers of genomes. The thickness of the curves represents the 99% confidence interval.

7. The 321 tomato lines used in GWAS only include modern, heirloom, and wild accessions of the tomato (*Solanum lycopersicum*) and its closest relative, *S. pimpinellifolium*. Please combine the reported panSV-genome with the SV identified here to generate a merged graph-based genome. Given that most of this set of phenotypic data are derived from modern varieties, with only one or two closely-related wild species, I doubt that these data sets are suitable to be used to assess the value of your pan-genome.

Response: Thanks for the suggestions. We have constructed a graph-based genome by integrating 358,543 SVs (those identified in this study and previously reported in Along *et al.*¹). SV genotyping by leveraging the graph-based genome indicated that a total of 164,170 SVs (45.8%) were polymorphic among the 321 tomato lines. Among them, 40,085 (24.4%) originated from the four accessions of SLL, SLC and SP in this study, 75,290 (45.9%) were from the eight other wild tomato species in this study, and the remaining 48,795 (29.7%) were derived from the inclusion of SVs reported in Along *et al.* These suggest that the 321-line tomato collection, albeit mostly being composed of accessions belonging to *S. lycopersicum* or *S. lycopersicum* var. *cerasiforme*, still contain a proportion of SVs derived from the

distantly related wild tomato species sampled in this study. In addition, this tomato collection is, however, the only one coupling with precisely documented phenotypic data. Therefore, these datasets would be sufficiently suitable to evaluate the value of our pan-genome, given the currently available data.

8. All codes and scripts used in this study should be deposited into a public platform or website, such as Github.

Response: We have deposited all codes and scripts used in bioinformatics analyses of this study on GitHub (<https://github.com/HongboDoll/TomatoSuperPanGenome>). This information has been added into the **Code availability** section.

References

1. Alonge, M. *et al.* Major Impacts of Widespread Structural Variation on Gene Expression and Crop Improvement in Tomato. *Cell* **182**, 145–161.e23 (2020).
2. Gao, L. *et al.* The tomato pan-genome uncovers new genes and a rare allele regulating fruit flavor. *Nat Genet* **51**, 1044–1051 (2019).
3. Wu, T. D. & Watanabe, C. K. GMAP: a genomic mapping and alignment program for mRNA and EST sequences. *Bioinformatics* **21**, 1859–1875 (2005).
4. Emms, D. M. & Kelly, S. OrthoFinder: phylogenetic orthology inference for comparative genomics. *Genome Biol.* **20**, 1–14 (2019).
5. Stein, J. C. *et al.* Genomes of 13 domesticated and wild rice relatives highlight genetic conservation, turnover and innovation across the genus *Oryza*. *Nat. Genet.* **50**, 285–296 (2018).
6. Gamuyao, R. *et al.* The protein kinase *Pstoll* from traditional rice confers tolerance of phosphorus deficiency. *Nature* **488**, 535–9 (2012).
7. Zhang, Z. *et al.* Genome-wide mapping of structural variations reveals a copy number variant that determines reproductive morphology in cucumber. *Plant Cell* **27**, 1595–604 (2015).
8. Lin, T. *et al.* Genomic analyses provide insights into the history of tomato breeding. *Nat. Genet.* **46**, 1220–1226 (2014).
9. Li, H. & Durbin, R. Fast and accurate short read alignment with Burrows–Wheeler transform. *bioinformatics* **25**, 1754–1760 (2009).
10. Servant, N. *et al.* HiC-Pro: an optimized and flexible pipeline for Hi-C data processing. *Genome Biol.* **16**, 1–11 (2015).
11. Burton, J. N. *et al.* Chromosome-scale scaffolding of de novo genome assemblies based on chromatin interactions. *Nat Biotechnol* **31**, 1119–25 (2013).
12. Consortium, 100 Tomato Genome Sequencing *et al.* Exploring genetic variation in the tomato (*Solanum* section *Lycopersicon*) clade by whole-genome sequencing. *Plant J.* **80**, 136–148 (2014).
13. Wang, X. *et al.* Genome of *Solanum pimpinellifolium* provides insights into structural variants during tomato breeding. *Nat. Commun.* **11**, 5817 (2020).
14. Sato, S. *et al.* The tomato genome sequence provides insights into fleshy fruit evolution. *Nature* **485**, 635–641 (2012).
15. Wellenreuther, M. & Bernatchez, L. Eco-Evolutionary Genomics of Chromosomal Inversions. *Trends Ecol Evol* **33**, 427–440 (2018).

16. Huang, K. & Rieseberg, L. H. Frequency, Origins, and Evolutionary Role of Chromosomal Inversions in Plants. *Front Plant Sci* **11**, 296 (2020).
17. Garrison, E. *et al.* Variation graph toolkit improves read mapping by representing genetic variation in the reference. *Nat Biotechnol* **36**, 875–879 (2018).
18. Ameur, A. Goodbye reference, hello genome graphs. *Nat Biotechnol* **37**, 866–868 (2019).
19. Strickler, S. R. *et al.* Comparative genomics and phylogenetic discordance of cultivated tomato and close wild relatives. *PeerJ* **3**, e793 (2015).
20. Lucatti, A. F., van Heusden, A. W., de Vos, R. C., Visser, R. G. & Vosman, B. Differences in insect resistance between tomato species endemic to the Galapagos Islands. *BMC Evol Biol* **13**, 175 (2013).
21. Sahu, K. K. & Chattopadhyay, D. Genome-wide sequence variations between wild and cultivated tomato species revisited by whole genome sequence mapping. *BMC Genomics* **18**, 430 (2017).
22. Chakraborty, M., Emerson, J. J., Macdonald, S. J. & Long, A. D. Structural variants exhibit widespread allelic heterogeneity and shape variation in complex traits. *Nat. Commun.* **10**, 1–11 (2019).
23. Goel, M., Sun, H., Jiao, W.-B. & Schneeberger, K. SyRI: finding genomic rearrangements and local sequence differences from whole-genome assemblies. *Genome Biol.* **20**, 277 (2019).

Decision Letter, first revision:

11th Aug 2022

Dear Professor Yu,

Your Article, "Super-pangenome Reveals the Panorama of Evolution and Structural Variation across Wild and Cultivated Tomato Species" has now been seen by 3 referees. You will see from their comments below that while they find your work of interest, some important points are raised. We are interested in the possibility of publishing your study in Nature Genetics, but would like to consider your response to these concerns in the form of a revised manuscript before we make a final decision on publication.

We therefore invite you to revise your manuscript taking into account all reviewer and editor comments. Please highlight all changes in the manuscript text file. At this stage we will need you to upload a copy of the manuscript in MS Word .docx or similar editable format.

*2) If you have not done so already please begin to revise your manuscript so that it conforms to our Article format instructions, available [here](http://www.nature.com/ng/authors/article_types/index.html). Refer also to any guidelines provided in this letter.

[redacted]

We hope to receive your revised manuscript within four to eight weeks. If you cannot send it within this time, please let us know.

Sincerely,
Wei

Wei Li, PhD
Senior Editor
Nature Genetics
New York, NY 10004, USA
www.nature.com/ng

Reviewers' Comments:

Reviewer #1:

Remarks to the Author:

Many thanks for the opportunity to read the revised manuscript. Most of my comments have been fully addressed. In particular the website is much improved in terms of speed, functionality and content.

The similarity in variant density between genic and intergenic regions is unusual given the fact that variation in genic regions is more likely to impact function and so would be selected against. This requires some clarification/comment.

While the majority of the text is well written, there remain some sections that require editing by a native speaker for clarity and the removal of hyperbole.

Examples include:

Line 57, reference genomes are clearly not 'critical' for plant breeding given that plant breeding was effective before genome sequencing became available.

Line 69, 'similar' to what?

Reviewer #2:

Remarks to the Author:

The revised manuscript has appropriately responded my concerns.

However, I have one more concern about the new data in this version. I do not think that the 9,394 genes were "really" uniquely present in this study, because the authors simply determined that based on the comparisons of gene coding sequence using the paramater ($< 90\%$ or coverage $< 75\%$). It was worthy to further analysis how many the so called unique genes were just a new alleles of genes presented in reported panels, or how many these unique genes in this study were arose because of un-collinear fragments. What the distribution of the unique genes, how many genes were expressed? Moreover, based on my experience, the coding sequence of a number of "new" genes were very short, and possibly unreliable. Therefore, I think that the author needs more careful analysis about that.

Reviewer #3:

Remarks to the Author:

In this revised version of the manuscript, authors sequenced and assembled chromosome-scale genomes of nine wild species and two cultivated accessions of tomato, and constructed a panSV-genome. They claimed that this study potentially provides valuable resources for tomato functional studies and breeding. The authors have done a lot of work to respond to the concerns raised by the previous reviewers. Although the manuscript has been significantly improved, I have the following concerns need to be addressed.

1. The quality of Supplementary Fig. 2 needs to be improved. The texts in this picture are blurred. I can't see the texts clearly even if I enlarge the picture.
2. Line 138-145, please indicate where to view the method used here.
3. Line 155-158, what is the criteria for determining whether a transposon is active or not? Why only indicate the Gypsy LRT-RTs in genomes of *S. lycopersicoides*, *S. chilense*, *S. peruvianum* are still active? How about *Copia*? Please indicate the method that you used to estimated insertion times here.
4. The quality of Supplementary Fig. 4 also needs to be improved. The texts in this picture also are blurred.
5. Line 177-180, 'phylogenetic analyses of chromosomes 1, 2, 9 and 11 showed *S. pennellii* was sister to other tomato species';
Line 170-171, 'Clade I encompassed two species: *S. pennellii* and *S. habrochaites*, and they are sister to other wild tomatoes'
The two sentences are so confusing. This expression makes the meaning of 'sister' confusing, and makes people misunderstand the meaning of phylogenetic trees.
6. The order of Fig. 1a and Fig. 1b should be reversed, since the text order has been changed.
7. Line 239-246, the authors find that several genes carried clade-specific variants, which are the evidence that prove the value of their data. The author should explore these genes further. For example, whether there are differences in the expression of these genes in different clade materials? What are the corresponding phenotypes of different clade wild tomato materials?
8. Line 259-262, please show the list of these fixed InDels.
9. Line 270-275, please add graph to show this result.
10. Line 314, The spelling of 'ref-fruited' seems to be wrong, which should be 'red-fruited'?
11. Line 334-335, what are the functions of the two genes that are labeled in Supplementary Fig. 21c. Please add this information in the manuscript.
12. Line 340-343, I can not draw this conclusion from this section. What kind of SVs in distantly wild tomatoes do you defined as those without artificial selection? Which genes are enriched in these SV without artificial selection?
13. Line 356-361, please show the structure of *Sgal12g015720* in all the wild tomato genomes that you reported in this manuscript.
14. Can these SVs in the last part only be detected using your SV data or pan-genome? Can they be identified by other pangenomes of tomato (Zhou et al, Nature.2022; Gao et al., Nat Genet. 2019, 51(6):1044-1051.; Alonge et al., Cell. 2020, 182(1):145-161.)? Do these SVs only exist in these distantly wild tomatoes?

Author Rebuttal, first revision:

Dear Dr. Wei Li,

We greatly appreciate your dedication and that of the reviewers to help us improve the manuscript.
Below we provide point-by-point responses to the reviewers' comments and indicate how we have
modified the manuscript. All revisions regarding reviewer's concerns have been highlighted in
**yellow** background.

Sincerely,

Qinghui Yu, on behalf of all co-authors who agreed on this submission

Reviewers' Comments:

Reviewer #1:

Remarks to the Author:

Many thanks for the opportunity to read the revised manuscript. Most of my comments have been
fully addressed. In particular the website is much improved in terms of speed, functionality and
content.

Response: Thanks for the comments.

The similarity in variant density between genic and intergenic regions is unusual given the fact that
variation in genic regions is more likely to impact function and so would be selected against. This
requires some clarification/comment.

Response: We totally agree with the reviewer that variants in gene coding regions are more likely
to undergo stronger positive selection. We thus checked the distribution of variants in genic regions
and found that the majority of these variants (81.9%) were localized in non-coding sequences
including introns and UTRs, with a density of 0.073 variants per bp. The remaining (18.1%) in
coding regions has a density of 0.039 variants per bp. To be clearer, we have rephrased
corresponding description to "Most of these variants (81.7%) were localized in intergenic regions
(0.063 variants per bp), whereas 3.3% resided within gene coding regions (0.039 variants per bp)".

While the majority of the text is well written, there remain some sections that require editing by a
native speaker for clarity and the removal of hyperbole.

Response: Thanks for the advice. We have carefully revised the manuscript with the aid of a native
speaker.

Examples include:

Line 57, reference genomes are clearly not 'critical' for plant breeding given that plant breeding was

effective before genome sequencing became available.

Response: Thanks for pointing this out. We have rephrased this sentence to “The availability of the
tomato reference genome (*S. lycopersicum* cv. Heinz 1706) has enabled comprehensive
characterization of genetic diversity in terms of ...” in the revised manuscript.

Line 69, ‘similar’ to what?

Response: Thanks for pointing this out. We have rephrased this sentence to “Recent advances in
tomato pan-genomics include”.

Reviewer #2:

Remarks to the Author:

The revised manuscript has appropriately responded my concerns.

However, I have one more concern about the new data in this version. I do not think that the 9,394
genes were “really” uniquely present in this study, because the authors simply determined that based
on the comparisons of gene coding sequence using the parameter ($< 90\%$ or coverage $< 75\%$). It
was worthy to further analysis how many the so called unique genes were just a new alleles of genes
presented in reported panels, or how many these unique genes in this study were arose because of
un-collinear fragments. What the distribution of the unique genes, how many genes were expressed?
Moreover, based on my experience, the coding sequence of a number of “new” genes were very
short, and possibly unreliable. Therefore, I think that the author needs more careful analysis about
that.

**Response:** Thanks for the helpful advice! In the revised manuscript, we have rephrased this section
and performed additional analyses to investigate these genes. Our analyses indicated that among the
9,320 non-redundant genes absent in the reported tomato pan-genome¹ (**Supplementary Table 14**),
5,158 arose due to non-collinear segments, and the remaining 4,162 were considered as additional
alleles of existing genes. These genes displayed a similar distribution pattern across the *S.*
*galapagense* genome, as compared with the whole-genome protein-coding genes (**Response Fig.**
**1a-c**), and their coding length was significantly shorter than the total genes predicted in the 13
tomato genomes (911 bp versus 1,370 bp, $p < 2.2 \times 10^{-16}$, Kruskal Wallis test), with a lower exon
number per gene (4 versus 5; **Response Fig. 1d,e**). We also observed that 69.3% (6,455 out of 9,320)
of these genes exhibited low expression (transcripts per million [TPM] ≤ 0.5 ; **Response Fig. 1f**).
These results have been integrated into the revised manuscript (**Lines 217–225**) and the
corresponding methods have also been added (**Lines 606–616**).

**Response Fig. 1. Features of the 9,320 genes identified in this study.** a) Karyotype of the 12
 tomato chromosomes using *S. galapagense* as the reference genome. b) Genome-wide distribution
 heatmap of the 9,320 genes identified in this study. c) Genome-wide distribution heatmap of the
 32,656 genes predicted in the *S. galapagense* reference genome. d) Distribution of coding sequence
 (CDS) length of genes predicted in all the 13 tomato genomes, the 4,162 genes being additional
 alleles of existing genes (Allelic) and the 5,158 genes arose due to non-collinear segments (Non-
 allelic). Only genes with CDS ≤ 7.5 kb were considered. e) Number of exons per gene in genes
 predicted in all the 13 tomato genomes, allelic and non-allelic genes (only genes carrying exons
 fewer than 30 were considered). In **d** and **e**, the 25% and 75% quartiles are shown as lower and
 upper edges of boxes, respectively, and central lines denote the median. The whiskers extend to 1.5
 85 times of the inter-quartile range. Data beyond the end of the whiskers are displayed as outliers. *P*-
 86 values were computed using the Kruskal-Wallis test. f) Percentage of genes predicted in all the 13
 tomato genomes, allelic and non-allelic genes showing high (transcripts per million (TPM) > 0.5)
 and low (TPM ≤ 0.5) levels of expression. *** *p*-values < 0.001 in Fisher's exact test.

Reviewer #3:

Remarks to the Author:

In this revised version of the manuscript, authors sequenced and assembled chromosome-scale
genomes of nine wild species and two cultivated accessions of tomato, and constructed a panSV-
genome. They claimed that this study potentially provides valuable resources for tomato functional
studies and breeding. The authors have done a lot of work to respond to the concerns raised by the
previous reviewers. Although the manuscript has been significantly improved, I have the following
concerns need to be addressed.

1. The quality of Supplementary Fig. 2 needs to be improved. The texts in this picture are blurred.
I can't see the texts clearly even if I enlarge the picture.

**Response:** Thanks for the suggestion. We have provided high-resolution images for Supplementary
Fig. 2.

2. Line 138-145, please indicate where to view the method used here.

**Response:** Thanks for the suggestion. We have added an extra description '**Methods, gene**
**prediction and functional annotation**' in bracket after the sentence "We combined *ab initio*
prediction, homology search, and transcriptome mapping approaches for protein-coding gene
prediction".

3. Line 155-158, what is the criteria for determining whether a transposon is active or not? Why
only indicate the Gypsy LTR-RTs in genomes of *S. lycopersicoides*, *S. chilense*, *S. peruvianum* are
still active? How about *Copia*? Please indicate the method that you used to estimated insertion times
here.

**Response:** We are sorry for the confusion. We initially focused on *Gypsy* LTR-RTs, since they
represent the most abundant type of LTR-RTs in the 13 tomato genomes. Since it is not appropriate
to determine whether a transposon is active solely from its insertion time, we have rephrased this
sentence to "Interestingly, we observed recent amplification of *Gypsy* and *Copia* LTR-RTs in four
wild tomato species (*S. lycopersicoides*, *S. corneliomulleri*, *S. peruvianum* and *S. chilense*;
**Supplementary Fig. 3**), implying that these wild species may have increasing degrees of genomic
diversity and environmental adaptability compared with cultivated tomatoes". We have also added
details regarding the estimation of LTR-RT insertion times in the **Method** section (**Lines 538–541**).

4. The quality of Supplementary Fig. 4 also needs to to be improved. The texts in this picture also
are blurred.

**Response:** Thanks for the suggestion. We have provided high-resolution images for Supplementary
Fig. 4 and also increased the font size.

5. Line 177-180, 'phylogenetic analyses of chromosomes 1, 2, 9 and 11 showed *S. pennelii* was
sister to other tomato species';

Line 170-171,' Clade I encompassed two species: *S. pennellii* and *S. habrochaites*, and they are
sister to other wild tomatoes'

The two sentences are so confusing. This expression makes the meaning of 'sister' confusing, and
makes people misunderstand the meaning of phylogenetic trees.

Response: We are sorry for the confusion. For the first sentence, we have rephrased it to “Clade I
encompassed two species, *S. pennellii* and *S. habrochaites*, which diverged from the common
ancestor of the other wild and cultivated tomatoes (except *S. lycopersicoides*) ~1.97 MYA.” For the
second one, we have rephrased it to “For example, within *Lycopersicon*, phylogenetic analyses with
genes from chromosomes 1, 2, 9 and 11 showed that *S. pennellii* was sister to other wild and
cultivated tomato species”.

6. The order of Fig. 1a and Fig. 1b should be reversed, since the text order has been changed.

Response: Thanks for pointing this out. We have reordered Fig.1a and Fig. 1b.

7. Line 239-246, the authors find that several genes carried clade-specific variants, which are the
evidence that prove the value of their data. The author should explore these genes further. For
example, whether there are differences in the expression of these genes in different clade materials?
What are the corresponding phenotypes of different clade wild tomato materials?

Response: Thanks for the suggestion. We fully agree with the Reviewer that it will be interesting
and valuable if these clade-specific variants are associated with phenotypes or gene expression of
different clades of wild tomatoes. Therefore, we have investigated expression patterns of these genes
under different haplotypes among the 13 tomato accessions (**Response Figs. 2-7**). However, in this
small panel, we failed to associate variations with corresponding expression levels, possibly because
the transcriptomic data used for quantifying gene expression were generated from whole-plant
mixed tissues, rather than specific tissues.

At current stage, we only have assembled genomes for ten tomato wild species, lacking
population-scale genomic sequences for these wild species. While some of the mentioned functional
genes possess direct impact on certain tomato traits, their functions have not yet been validated in
wild tomato species. Therefore, we think it is currently inconvincible to draw conclusions or to
make speculations for phenotypic outcome of these variants, which is also beyond the scope of this
study. Associations of these variants with phenotypes and their functional characterizations in
different clades of tomato wild species could be conducted in future research, once genome
resequencing data for a sufficiently large panel of individuals are available. We appreciate your
understanding.

Response Fig. 2. Haplotypes of *Sgal01g012480* among genomes of *S. lycopersicoides* and the 12 tomatoes. Clade IV species have the same haplotype except one C/T variation on the first exon of this gene in *S. lycopersicum* var. *cerasiforme*. The G/A allele at the first exon clearly separates clade IV species from other accessions. Sanger sequencing results for certain SNPs are also shown. Expression fold-changes of *Sgal01g012480* haplotypes of the 12 tomato accessions, compared with the *S. galapagensis* haplotype are displayed using bar plots. Positive fold-changes denote increased expression, whereas minus fold-changes indicate reduced expression.

Response Fig. 3. Haplotypes of *Sgal01g016480* among genomes of *S. lycopersicoides* and the 12 tomatoes. Except for *S. galapagense*, other three genomes in clade IV have the same allele at four positions (sequenced two), which might be one of the evidences that cultivated tomatoes may be domesticated from *S. pimpinellifolium*. Sanger sequencing results for certain SNPs are also shown. Expression fold-changes of *Sgal01g016480* haplotypes of the 12 tomato accessions, compared with the *S. galapagense* haplotype are displayed using bar plots. Positive fold-changes denote increased expression, whereas minus fold-changes indicate reduced expression.

Response Fig. 4. Haplotypes of *Sgal01g019530* among genomes of *S. lycopersicon* and the 12 tomatoes. The C/G, C/A, G/A alleles clearly separate clade IV species from other wild relatives. Moreover, C/T and A/T alleles in the 5th and the 10th exons distinguish M82 and Heinz1706 to other species. Sanger sequencing results for certain SNPs are also shown. Expression fold-changes of *Sgal01g019530* haplotypes of the 12 tomato accessions, compared with the *S. galapagensis* haplotype are displayed using bar plots. Positive fold-changes denote increased expression, whereas minus fold-changes indicate reduced expression.

**Response Fig. 5. Haplotypes of *Sgal01g019540* among genomes of *S. lycopersicoides* and the**
 **12 tomatoes. One *S. galapagensis*-specific allele exists at the 11th exon. Sanger sequencing results**
 **for the SNP are also shown. Expression fold-changes of *Sgal01g019540* haplotypes of the 12 tomato**
 **accessions, compared with the *S. galapagensis* haplotype are displayed using bar plots. Positive**
 **fold-changes denote increased expression, whereas minus fold-changes indicate reduced expression.**

Response Fig. 6. Haplotypes of *Sgal02g028860* among genomes of *S. lycopersicoides* and the 12 tomatoes. Except for *S. pimpinellifolium*, other species in clade IV have the same haplotype. Sanger sequencing results for the SNP are also shown. Expression fold-changes of *Sgal02g028860* haplotypes of the 12 tomato accessions, compared with the *S. galapagensis* haplotype are displayed using bar plots. Positive fold-changes denote increased expression, whereas minus fold-changes indicate reduced expression.

Response Fig. 7. Haplotypes of *Sgal03g023990* among genomes of *S. lycopersicoides* and the 12 tomatoes. Two *S. galapagense* specific allele are localized at the first and second exons. Sanger sequencing results for certain SNPs are also shown. Expression fold-changes of *Sgal03g023990* haplotypes of the 12 tomato accessions, compared with the *S. galapagense* haplotype are displayed using bar plots. Positive fold-changes denote increased expression, whereas minus fold-changes indicate reduced expression.

8. Line 259-262, please show the list of these fixed InDels.
Response: Thanks for the suggestion. We have added a Supplementary Table to list these variants
(Supplementary Table 19).

9. Line 270-275, please add graph to show this result.
Response: Thanks for the suggestion. We have added a Supplementary Figure to visualize these
results (Supplementary Fig. 21).

10. Line 314, The spelling of ‘ref-fruited’ seems to be wrong, which should be ‘red-fruited’?
Response: We are sorry for the mis-spelling. We have corrected it to “red-fruited” in the revised
manuscript.

11. Line 334-335, what are the functions of the two genes that are labeled in Supplementary Fig.
21c. Please add this information in the manuscript.
Response: Thanks for the suggestion. We have added additional functional description for
*Sgal12g015720* in the revised manuscript (Lines 355–358). Since we cannot assign any functional
annotation to *Sgal08g000320*, we have removed the corresponding label in the Supplementary
Figure.

12. Line 340-343, I can not draw this conclusion from this section. What kind of SVs in distantly
wild tomatoes do you defined as those without artificial selection? Which genes are enriched in
these SV without artificial selection?
Response: We are sorry for confusion. We have rephrased this sentence to “These results suggest
that SVs in these distantly related wild tomatoes have undergone distinct evolutionary trajectories
than cultivated tomatoes and their progenitors”, since we cannot infer whether an SV has undergone
“artificial” selection from currently available data.

13. Line356-361, please show the structure of Sgal12g015720 in all the wild tomato genomes that
you reported in this manuscript.
Response: Thanks for the suggestion. We have added a Supplementary Figure to visualize the
structure of this gene in the nine wild tomato species (Supplementary Fig. 24).

14. Can these SVs in the last part only be detected using your SV data or pan-genome? Can they be
identified by other pangomes of tomato (Zhou et al, Nature.2022; Gao et al., Nat Genet. 2019,
51(6):1044-1051.; Alonge et al., Cell. 2020, 182(1):145-161.)? Do these SVs only exist in these
distantly wild tomatoes?
Response: Some of these SVs can also be found in other tomato pan-genome studies. In the final
section of the manuscript, we used 360,189 SVs (224,447 SVs identified in this study and 135,742
SVs previously reported in Alonge *et al.*²) to conduct GWAS. Regarding the identified peak SVs,

we observed that 168 out of the 714 (23.5%) peak SVs could be found in Alonge *et al.*², and 135
out of the 714 (18.9%) peak SVs could be recovered in Zhou *et al.*³. Since analyses of Gao *et al.*¹
were based on resequencing short reads and did not provide a catalog of SVs, we cannot make the
comparison feasible. Among the 714 identified peak SVs, 85 were only present in these distantly
wild tomato species used in this study.

**References**

- 1. Gao, L. *et al.* The tomato pan-genome uncovers new genes and a rare allele regulating fruit
flavor. *Nat Genet* **51**, 1044–1051 (2019).
- 2. Alonge, M. *et al.* Major impacts of widespread structural variation on gene expression and
crop improvement in tomato. *Cell* **182**, 145-161.e23 (2020).
- 3. Zhou, Y. *et al.* Graph pangenome captures missing heritability and empowers tomato
breeding. *Nature* **606**, 527–534 (2022).

Decision Letter, second revision:

4th Nov 2022

Dear Dr. Yu,

Thank you for submitting your revised manuscript "Super-pangenome Reveals the Panorama of Evolution and Structural Variation across Wild and Cultivated Tomato Species" (NG-A58997R1). It has now been seen by the original referees and their comments are below. The reviewers find that the paper has improved in revision, and therefore we'll be happy in principle to publish it in Nature Genetics, pending minor revisions to satisfy the referees' final requests and to comply with our editorial and formatting guidelines.

Sincerely,
Wei

Wei Li, PhD
Senior Editor
Nature Genetics
New York, NY 10004, USA
www.nature.com/ng

Reviewer #1 (Remarks to the Author):

Many thanks for the clarification regarding the variant density. The manuscript is improved, though the use of English could be refined further. I expect this could happen at the proofing stage. I have no further suggestions.

Reviewer #2 (Remarks to the Author):

The revised manuscript has appropriately responded my major concerns. However, there was one point which the author did not get. About the distribution of the unique genes, obviously, I mean the distribution of unique gene in tomato accession population, not along chromosome, then it make sense.

Reviewer #3 (Remarks to the Author):

In the revised version of the ms, authors have fully addressed all my comments.

Author Rebuttal, second revision:

Dear Dr. Wei Li,

We greatly appreciate your dedication and that of the reviewers to help us improve the manuscript. Below we provide point-by-point responses to the reviewers' comments and indicate how we have modified the manuscript. All revisions regarding reviewer's concerns have been highlighted in **yellow** background.

Sincerely,

Qinghui Yu, on behalf of all co-authors who agreed on this submission

Reviewers' Comments:

Reviewer #1 (Remarks to the Author):

Many thanks for the clarification regarding the variant density. The manuscript is improved, though the use of English could be refined further. I expect this could happen at the proofing stage. I have no further suggestions.

Response: Thanks for the comments.

Reviewer #2 (Remarks to the Author):

The revised manuscript has appropriately responded my major concerns. However, there was one point which the author did not get. About the distribution of the unique genes, obviously, I mean the distribution of unique gene in tomato accession population, not along chromosome, then it make sense.

Response: We are sorry for the confusion. In the revised manuscript, we have investigated the distribution of those unique genes in a tomato population comprising 321 accessions, and found that more than half of these genes are absent in all the 321 tomatoes (**Supplementary Note 3** and **Supplementary Fig. 9**).

Reviewer #3 (Remarks to the Author):

In the revised version of the ms, authors have fully addressed all my comments.

Response: Thanks for the comments.

Final Decision Letter:

21st Feb 2023

Dear Dr. Yu,

I am delighted to say that your manuscript "Super-pangenome analyses highlight genomic diversity and structural variation across wild and cultivated tomato species" has been accepted for publication in an upcoming issue of Nature Genetics.

Your paper will be published online after we receive your corrections and will appear in print in the next available issue. You can find out your date of online publication by contacting the Nature Press Office (press@nature.com) after sending your e-proof corrections. Now is the time to inform your Public Relations or Press Office about your paper, as they might be interested in promoting its publication. This will allow them time to prepare an accurate and satisfactory press release. Include your manuscript tracking number (NG-A58997R2) and the name of the journal, which they will need

when they contact our Press Office.

Please note that *Nature Genetics* is a Transformative Journal (TJ). Authors may publish their research with us through the traditional subscription access route or make their paper immediately open access through payment of an article-processing charge (APC). Authors will not be required to make a final decision about access to their article until it has been accepted. [Find out more about Transformative Journals](https://www.springernature.com/gp/open-research/transformative-journals)

Authors may need to take specific actions to achieve [compliance](https://www.springernature.com/gp/open-research/funding/policy-compliance-faqs) with funder and institutional open access mandates. If your research is supported by a funder that requires immediate open access (e.g. according to [Plan S principles](https://www.springernature.com/gp/open-research/plan-s-compliance)) then you should select the gold OA route, and we will direct you to the compliant route where possible. For authors selecting the subscription publication route, the journal's standard licensing terms will need to be accepted, including [self-archiving-and-license-to-publish](https://www.nature.com/nature-portfolio/editorial-policies/self-archiving-and-license-to-publish). Those licensing terms will supersede any other terms that the author or any third party may assert apply to any version of the manuscript.

Please note that Nature Portfolio offers an immediate open access option only for papers that were first submitted after 1 January, 2021.

If you have not already done so, we invite you to upload the step-by-step protocols used in this manuscript to the Protocols Exchange, part of our on-line web resource, [natureprotocols.com](https://www.nature.com/natureprotocols). If you complete the upload by the time you receive your manuscript proofs, we can insert links in your article that lead directly to the protocol details. Your protocol will be made freely available upon publication of your paper. By participating in [natureprotocols.com](https://www.nature.com/natureprotocols), you are enabling researchers to more readily reproduce or adapt the methodology you use. [Natureprotocols.com](https://www.nature.com/natureprotocols) is fully searchable, providing your protocols and paper with increased utility and visibility. Please submit your protocol to <https://protocolexchange.researchsquare.com/>. After entering your nature.com username and password you will need to enter your manuscript number (NG-A58997R2). Further information can be found at <https://www.nature.com/nature-portfolio/editorial-policies/reporting-standards#protocols>

Sincerely,
Wei

Wei Li, PhD
Senior Editor
Nature Genetics
New York, NY 10004, USA
www.nature.com/ng